environmental science/statistics/environmental engineering

degraded pasturelands, livestock efficiency, rural credit, Brazil

**Author for correspondence:**
Rafael Feltran-Barbieri
e-mail: rafael.barbieri@wri.org

# Degraded pastures in Brazil: improving livestock production and forest restoration

Rafael Feltran-Barbieri[1] and José Gustavo Féres[2,3]

[1]World Resources Institute Brazil, Rua Claudio Soares 72, Sala 1510, São Paulo SP 05422-030, Brazil
[2]Institute for Applied Economic Research (IPEA), Avenida Presidente Vargas 730, 17o Andar, Rio de Janeiro RJ 20071-900, Brazil
[3]Brazilian School of Economics and Finance (EPGE-FGV), Praia de Botafogo 190, 11o Andar, Rio de Janeiro RJ 22250-900, Brazil

RF-B, 0000-0002-9782-2486

Degraded pasture is a major liability in Brazilian agriculture, but restoration and recovery efforts could turn this area into a new frontier to both agricultural yield expansion and forest restoration. Currently, rural properties with larger degraded pasture areas are associated with higher levels of technical inefficiency in Brazil. The recovery of 12 million ha of degraded pastures could generate an additional production of 17.7 million bovines while reducing the need for new agricultural land. Regional identification of degraded pastures would facilitate the targeting of agricultural extension and advisory services and rural credit efforts aimed at fostering pasture recovery. Since only 1% of Brazilian municipalities contain 25% of degraded pastures, focusing pasture recovery efforts on this small group of municipalities could generate considerable benefits. More efficient allocation of degraded and native pastures for meat production and forest restoration could provide land enough to fully comply with its Forest Code requirements, while adding 9 million heads to the cattle inventory. Degraded pasture recovery and restoration is a win–win strategy that could boost livestock husbandry and avoid deforestation in Brazil and has to be the priority strategy of agribusiness sector.

## 1. Introduction

Global population is expected to reach 10 billion people in the next 30 years. The Food and Agriculture Organization projects that additional 52 million tons of nitrogen fertilizer and 165 million ha of new agricultural land will be required to meet global food, feed, fibre and biofuel demand by 2050. These variations

correspond to an increase of 50% in fertilizer use and 6% in agricultural land, compared with 2012 [1]. The simultaneous requirement for more fertilizers and land is explained not only by regional differences regarding production factor endowments, technology, and income and land tenure (LT) inequalities. Inefficiencies and misspending in supply processes are responsible for significant resource losses. Post-harvest output losses have been estimated to reach 1.3 billion tons annually [2]. In addition, 17 trillion tons of topsoil are lost every year worldwide, resulting in economic costs up to US$ 8 billion [3]. Erosion drives land abandonment and degradation. Globally, 1 billion ha of arable land are abandoned or degraded [4].

Inefficient agricultural management and growing food demand have promoted a sectoral expansion pattern, with severe environmental impacts. Agricultural expansion is recognized as a major deforestation driver in tropical regions around the world. Brazil, Indonesia, Republic of Congo, Colombia, Laos and Mozambique have lost 50 Mha of forests since 2001 [5].

In addition to biodiversity losses and socio-economic impacts on the local population, deforestation results in depreciation of ecosystem services necessary for agriculture. Climate stability, soil fertility, water availability and quality, pollination and biological pest control are essential conditions for assuring agricultural productivity. The Intergovernmental Panel on Climate Change [6] has highlighted four conditions for achieving food security: (i) producing more food in sites with shortage and scarcity, (ii) strengthening global and local governance in food supply, demand and accessibility, (iii) reducing deforestation and promoting forest restoration, and (iv) improving efficiency in food production.

Forest restoration and recovery of degraded lands are key strategies for achieving food security goals, and the Brazilian agricultural sector could play a leading role in this initiative. The country is an agricultural powerhouse, but it has also accumulated around 100 Mha of degraded pasturelands [7]. Implementing restoration and recovery actions would result in significant environmental and economic gains.

The Brazilian National Determined Contribution (NDC)—a commitment assumed as part of the Paris Agreement to mitigate and adapt the economy to climate change—recognizes that restoring forests and recovering degraded pasturelands are core strategies for reducing deforestation pressures. The Brazilian NDC has committed to recovering 15 million ha of degraded pasturelands, to restoring 12 million ha of native vegetation, and to creating 5 million ha of integrated crop–livestock–forest and silvopastoral systems by 2030.

To highlight the role that pasture recovery may play in achieving economic goals while improving forest restoration, this paper assesses the relationship between pasture quality, technical efficiency (TE) and livestock production.

## 1.1. Livestock in Brazil

Brazil produces 16% of the world's beef and accounts for 20% of the global beef market [8], having traded about US$ 7.6 billion Free on Board (FOB) [9] in 2019. One-third of the Brazilian agribusiness GDP, or about US$ 81 billion, is generated by cattle livestock, a sector that employs 3 million people in rural areas [10,11].

Despite its socio-economic importance, the Brazilian cattle sector has performed far below its biophysical potential. The observed average productivity is 89 kg ha$^{-1}$ yr$^{-1}$. However, biocapacity exceeds 172 kg ha$^{-1}$ yr$^{-1}$ [7]. Despite significant regional heterogeneity in technology adoption and production specialization, extensive and inefficient production systems are unfortunately common in the country.

Unlike crops such as soya beans and sugarcane, whose technologies provide little room for input substitution, production factors are more interchangeable in livestock activities. Specifically, land and capital present a high degree of input substitutability: farmers may use additional pastureland at the expense of capital. Conversely, rural producers may resort to intensive mechanization, with resulting productivity gains reducing the need for new agricultural areas. This explains the coexistence of ultra-extensive livestock farms in the Amazon and the high-tech feedlot systems in the Centre-South of Brazil. In addition to that, while cattle are a capital stock with high liquidity, land is a real asset that may be employed as a hedging strategy in periods of macroeconomic instability. These characteristics of cattle and land may reinforce the incentive to expand agricultural frontiers, adopting extensive, rather than intensive, practices [12].

In the last 35 years, around 45 Mha of new pastures were added to the Brazilian agricultural landscape [13]. During this time, a huge area of 64 Mha was deforested and converted to new pastures, while 18 Mha of pre-existing pastures have been replaced by agriculture, forestry and dams [13]. Currently, 70% (37 Mha) of total pastureland in the Amazon may be attributed to the deforestation process that took place in the last 35 years. At the same time, one-third of existing

pastures may be attributed to deforestation processes occurring in the Cerrado and Atlantic Forest biomes. Unfortunately, the expansion of the agricultural frontier at the expense of natural vegetation is still ongoing. Since 2010, 10 Mha of new pastures and 4 Mha of new croplands have expanded, replacing forests and other natural vegetation [13].

Meanwhile, the carcass weight produced increased by only 10% (0.74% $yr^{-1}$), and a slight gain in the stocking rate from 248 to 255 kg $ha^{-1}$ was observed over the last decade. These figures are in sharp contrast to the considerable productivity gains experienced in corn and soya bean production. Corn productivity grew by 5.3% $yr^{-1}$, while soybean productivity rose by 3.9% $yr^{-1}$. Both crops present productivity indicators comparable to Mercosur neighbours, and even the USA, while beef productivity is at least 20% less than that of the main competitors [11].

The relationship between deforestation pressures and extensive livestock production has raised concerns regarding access to export markets. Representatives of agribusiness have engaged in several initiatives to prevent deforestation, signalling a commitment to sustainable practices. The Brazilian Coalition on Climate, Forest and Agriculture (Coalizão Brasil, Clima, Florestas e Agricultura), a multi-sector agreement with 250 institutions, including the Brazilian Agribusiness Association (ABAG), Brazilian Beef Exporters Association (ABIEC) and three major commercial banks, has announced a proposal to halt deforestation. The initiative has adopted several measures, including tracing the production chain, and cutting investments in producers and industries found to be directly or indirectly responsible for illegal deforestation [14].

These pressures call into question the livestock growth model based on extensive practices and adding new agricultural lands. At the same time, they present an opportunity to boost alternative models such as recovery of degraded pastures. In the following sections, this article assesses potential benefits from degraded pasture recovery, and identifies some mechanisms to provide incentives for adopting such practices in Brazil. We are interested in pursuing the following specific objectives:

(1) to estimate the TE of livestock production in Brazil and to relate TE to pastureland characteristics (planted, native or degraded);
(2) to estimate the increase in Brazil's cattle production that could be achieved by recovering degraded pasturelands; and
(3) to identify priority sites and policies that should be starting points for a new livestock system.

Unlike previous studies which used GIS tools to identify and measure the impact of degraded pasture recovery [15–19], we focus on self-reported data regarding degraded pasture provided by the Agricultural Census [11]. We believe that this option is in line with a more positive and realistic approach to assess the potential economic gains from pasture recovery and restoration efforts. Farmers are more prone to undertake action in areas which they recognize as degraded, while they may be more reluctant to act in areas they do not consider to be in poor condition.

We also depart from the current practice of adopting the biophysical potential as the reference in terms of adding cattle. To assess economic gains, we use the regional stocking rate of non-degraded pastures locally estimated by spatial error model regressions (SEM). This procedure also provides a more realistic estimate for potentially enlarging the cattle herd, since it reflects the average stocking rate associated with technologies effectively adopted by local ranchers.

# 2. Material and methods

## 2.1. Degraded pastures

Degraded pasturelands are native or planted pastures which have experienced a sharp decrease in carrying capacity, productivity and biomass production. Degradation may result from inadequate soil, plant or herd management. Degradation is normally related to overgrazing, insufficient weed and pest controls, and lack of fertilization [16].

There are about 1.1 Gha of degraded lands worldwide [4]. In Brazil, the Atlas of Brazilian Pasturelands reports that 57% of the total 173 million hectares of pasturelands were degraded by 2018. More critically, the Atlas registered that approximately 40 Mha of pasture suffer from a severe level of degradation [7]. In contrast with this georeferenced estimate, rural producers recognize only 12 Mha of pasture to be in poor condition on their properties, as reported on the last Agricultural Census, conducted in 2017 [11].

Discrepancy between geoprocessing analysis and the producers' self-reported measure of degraded conditions of their own pasturelands is expected. The difference may have several causes. Producers may have different subjective perceptions about what they consider to be a degraded pasture. They may also have incentives to under-report degraded land area to avoid regulatory sanctions. We will not address the many causes underlying the discrepancy between georeferenced data and farmers' reported data.

For the purposes of our analysis, then, we consider the degraded pasture as reported by farmers. Producers' recognition of degraded pastures is a crucial step towards adopting more efficient and productive technologies [19,20]. The technical or empirical frameworks used by producers reflect their beliefs, knowledge, and local experiences and practices. Focusing on what was declared in the Census allows us to work in areas that farmers recognize as degraded, without any controversy. This is consistent with the decision-maker level relevant to technology adoption and our TE analysis. Therefore, we recognize the degraded pasture area from the rural producers' perspective, so we can treat our potential gains from recovering degraded pasture as a conservative estimate.

## 2.2. Technical efficiency

Producers can be characterized as efficient if they produce as much output as possible with the inputs they employ, or if they produce a certain output quantity at minimum cost. In our empirical application, we calculate efficiency using the concept of TE [21,22]. TE measures the gap between observed production and maximum potential production. Thus, we can define technical inefficiency by how far the observed output is from potential output. Technical inefficiency may be interpreted as the additional output the farmer could produce while using the current amount of input or, conversely, the potential reduction in input use that could be attained while producing the current output quantity.

Formally, TE may be expressed by the formula

$$\text{TE} = \frac{y}{\bar{y}} \leq 1, \tag{2.1}$$

where TE denotes technical efficiency, $y$ represents the observed output, in monetary value, and $\bar{y}$ is the potential (maximum) output. As $y$ tends to the potential output $\bar{y}$, TE increases and the producer becomes more efficient. We say that a farmer is efficient if TE = 1. Low TE values mean that observed production is far below the potential level, indicating higher inefficiency.

Several different approaches exist to estimate TE. We adopt the stochastic frontier analysis, based on econometrics. To derive an econometric model, we first observe that potential output $\bar{y}$ may be represented by the production function $f(x, \boldsymbol{\beta})$, where $x$ is a vector of input quantities, and $\boldsymbol{\beta}$ is a vector of parameters to be estimated. Equation (2.1) may be rewritten as

$$y = \bar{y}\,\text{TE} = f(x, \boldsymbol{\beta})\text{TE}. \tag{2.2}$$

We adopt a Cobb–Douglas specification for our production function. The empirical model for farm $i$ is linear in the logarithm of the variables and it can be expressed as

$$\ln(y_i) = = \ln f(x_i, \boldsymbol{\beta}) + \ln(\text{TE}_i) = \ln f(x_i, \boldsymbol{\beta}) - u_i, \tag{2.3}$$

where $u_i \geq 0$ is a measure of technical inefficiency, since $u_i = -\ln \text{TE}_i \approx 1 - \text{TE}_i$. We observe that

$$\text{TE}_i = -u_i. \tag{2.4}$$

Our input vector $x$ includes capital, labour and land. Capital is measured by the total quantity of tractors, agricultural machinery and equipment on all cattle farms, in each municipality. Labour corresponds to the total number of employees. Finally, land is the total area of pasture. Pastures are disaggregated into three components: degraded, native and planted pasturelands. This decomposition allows an assessment of TE associated with each type of pasture. In particular, we can evaluate how reducing degraded pasture areas could affect TE. Additional biophysical and agricultural variables are used as controls (see electronic supplementary material for details).

## 2.3. Productivity

Livestock productivity is a function of stocking rate, carcass weight equivalent and offtake rate. Commonly, production intensification is achieved by a simultaneous gain in all three parameters.

Productivity is increased by a combination of several factors, including proper choice of breeds, pasture management, and maintaining health and comfort of the animals [16]. Improved pasture productivity can be achieved by other practices. Conventional practices include soil correction, chemical or organic fertilization, and soil conservation by contour lines, while innovative management extends to crop–livestock integration and silvopastoral systems [23].

Numerous studies conducted in Latin America and Brazil show how integrated systems could raise livestock productivity, scale-up production and promote income diversification [24–27]. Additional benefits include animal welfare and climate change adaptation and mitigation [28,29]. The Brazilian NDC recognizes the importance of these systems, with a goal of creating 5 million ha of integrated systems by 2030. Brazil already has between 8 and 12 million ha involved in crop–livestock integration, and another 1–1.5 million ha in crop–livestock–forest, in over 70 different systems [30].

With such a diversity of integrated systems, any evaluation of the potential to increase production on a national scale faces major challenges. Furthermore, conventional systems of non-integrated production and traditional management in pasture recovery—e.g. soil correction, fertilization, contour lines and grass seeding—are still dominant in Brazil, adding up to a total area of approximately 170 Mha [16]. The most common management in the region would be captured implicitly by the coefficients of regressions of the stocking rate. In what follows, we restrict our analysis to conventional systems.

## 2.4. Stocking rates

We estimate the additional herd numbers associated with the recovery of degraded pasturelands as a function of pasture area and productivity. As described in §2.1, the degraded pasture area is based on farmers' self-declaration. For the potential productivity, unlike previous studies that consider the biophysical potential [17,31], we adopt the average current stocking rate of non-degraded pasturelands.

Non-degraded pastures may be either native or planted pasturelands with distinct stocking rates, due to natural and management attributes. The estimated stocking rate for each type of pasture provides the current level that degraded pasture recovery could achieve, given the current available technology. Since we are more interested in a positive, rather than a normative, perspective on possible gains of livestock intensification, we work with the observed productivity of each type of pasture, rather than their biophysical potential. This positive perspective also aligns with the concept of TE.

Therefore, our results should be interpreted as conservative. There would be plenty of room to recover larger areas of degraded pastures, and to gain even higher productivity, thanks to technological advances. This supports the technical and economic viability of the pasture recovery scenarios we present here.

Pastures are explicitly spatial variables, and testaments to local landscape management. To evaluate the effect of production factors (and, in particular, different pasture types) on cattle stocking rates, we apply the spatial error model. This model adequately treats spatial dependence in the error term, an issue that arises from unobservable latent variables that are spatially correlated [32]. Formally, the SEM may be expressed by the formula

$$Y = \mathbf{X}\boldsymbol{\beta} + \varepsilon$$
$$\varepsilon = pW\varepsilon + u, \tag{2.5}$$

where $Y$ denotes the dependent variable, $\mathbf{X}$ is the matrix of independent variables, $\boldsymbol{\beta}$ represents the corresponding parameter vectors of $\mathbf{X}$, $\varepsilon$ is the error term, $p$ is the spatial scalar disturbance, $W$ stands for the spatial-weighting matrix and $u$ is the random error. The $pW\varepsilon$ is the autoregressive component of the error term.

Assuming

$$|p| < 1, \ \varepsilon = (I - pW)^{-1}u. \tag{2.6}$$

Equation (2.5) may be described as

$$Y = \mathbf{X}\boldsymbol{\beta} + (I - pW)^{-1}u, \tag{2.7}$$

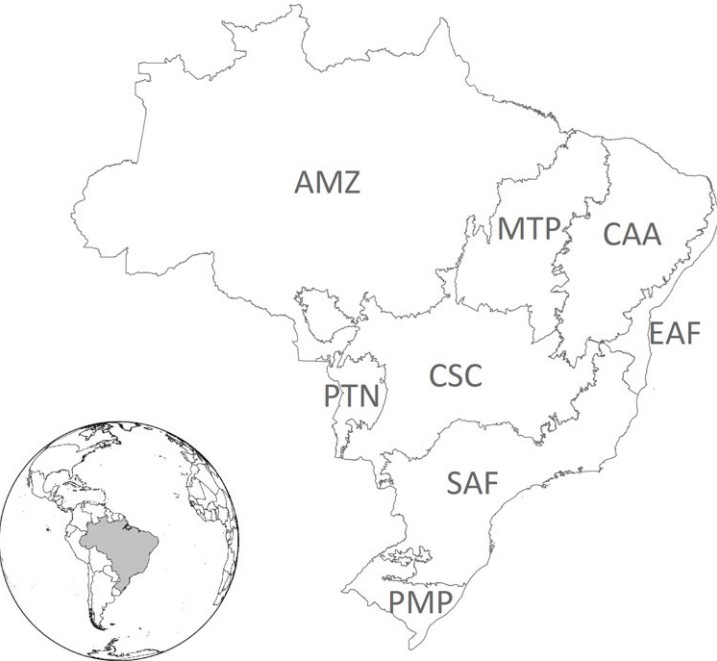

**Figure 1.** Brazilian regions. Amazon (AMZ), Matopiba Cerrado (MTP), Caatinga (CAA), Eastern Atlantic Forest (EAF), Pantanal (PTN), Centre-Southern Cerrado (CSC), Southern Atlantic Forest (SAF) and Pampa (PMP).

with $W$ the row-standardized spatial weight matrix built from an inverse of distance among all sample municipalities (km)

$$W = \begin{bmatrix} 0 & \frac{1}{d_{1,2}} & \cdots & \frac{1}{d_{1,j-1}} & \frac{1}{d_{1,j}} \\ \frac{1}{d_{2,1}} & 0 & \cdots & \frac{1}{d_{2,j-1}} & \frac{1}{d_{2,j}} \\ \vdots & \vdots & \ddots & \vdots & \vdots \\ \frac{1}{d_{i-1,1}} & \frac{1}{d_{i-1,2}} & \cdots & 0 & \frac{1}{d_{i-1,j}} \\ \frac{1}{d_{i,1}} & \frac{1}{d_{i,2}} & \cdots & \frac{1}{d_{i,j-1}} & 0 \end{bmatrix},$$ (2.8)

where $d_{i,j}$ is the distance between municipality $i$ and municipality $j$.

In our models, Y represents the total cattle herd on all farms in a given municipality, while the matrix **X** includes the, respectively, degraded, native and planted pasture areas and other control variables. Notably, SEM estimation suffers from a misspecification, and the results of the model are biased, if no spatial dependence is identified. In this case, robust regression is applied for the same set of observations, as recommended by the authors in [32,33]. We apply final specifications for eight regions, as described in figure 1.

In these specifications, the difference between the coefficients for degraded pasture and planted pasture is assumed to be the impact of pasture recovery. The underlying assumption is that the increase in cattle herd size that could be achieved from pasture recovery is a function of the total area recovered and differences in stocking rates among the pasture types, all else being constant.

We consider two scenarios: (i) a high-intensification scenario (HIS), in which all degraded pasture is recovered, and achieves the stocking rate estimated for planted pasturelands, and (ii) Forest Code Compliance (FCC), which allocates degraded pastures for forest restoration to eliminate the deficit of forest cover, as mandated by the Forest Code (FC). Remaining degraded pasture areas, exceeding the requirements to comply with the FC, if any, are recovered and achieve the stocking rate estimated for planted pasturelands. Native pastures are allocated to cover remaining areas to be restored, if any.

For this scenario, it is important to note that the FC (Law 12651/12) mandates the conservation or restoration of the Permanent Preservation Areas (PPA), which comprise water-, soil- and biodiversity-sensitive sites as gallery forests; and the Legal Reserves (LR), in addition to PPAs, which must be allocated for low-impact activities such as selective logging. The size of LRs depends on biome and

farm size, ranging from 0% of total farm area in small farms all over the country to 80% in large farms in the Amazon, 35% in Cerrado in the Legal Amazon and 20% in other biomes and regions.

However, an estimated 2 million farms in Brazil (33% of the total) do not have enough PPA or LR to comply with the FC, adding up to a 'deficit' of between 18 and 21 Mha of native vegetation [34,35]. Brazilian legislation establishes that part of the LR deficit may be compensated for outside of farms, on the condition that the area of native vegetation is in the same state and the same biome in which the farm is located. On the other hand, Brazilian legislation is clear in stating that land planning is the responsibility of the federal government, but the implementation is delegated to municipalities. For the FCC, we target the municipality level for allocation, following the Agriculture Atlas produced by Imaflora [35]. This means that a region may have less degraded pastureland than vegetation deficit, although many municipalities have a surplus of pastureland. We consider the eight regions shown in figure 1.

# 3. Results and discussion

## 3.1. Efficiency, stocking rates and additional herd

We estimated TE using 4136 out of 5563 municipalities, due to missing values for 1527 municipalities. The sample encompasses 8.9 out of 10.0 Mha of degraded pastures on cattle farms. We estimated stocking rates and additional herd numbers in proposed scenarios using 4617 municipalities, and 10.8 out of the 11.8 Mha of degraded pastures on all farms.

Our stochastic frontier model estimates that Brazilian cattle ranching has an average TE of 0.81. This means that, on average, current production corresponds to 81% of the potential level of production. In other words, the current value of agricultural production could be increased by approximately 23% using the same amount of inputs. Improvements may come from better cattle management, pasture rotation, water accessibility, better planning of expenses and investments, and, mainly, by reducing waste of inputs and unnecessary operations.

Thus, TE measures how much a system could be improved rather than how good it is compared with others [36]. For comparison, productivity or yield ($Y$) is a better measure. For instance, in Nigeria TE of large livestock farms is 83% with $Y$ of an average of 45 kg ha$^{-1}$ yr$^{-1}$ [36], while TE in grass-fed livestock systems in the USA is 84% [37], but has an average $Y$ of hugely 141 kg ha$^{-1}$ yr$^{-1}$ [38]. On the other hand, TE in Uruguay is 77% with $Y$ of 95 kg ha$^{-1}$ yr$^{-1}$ [39], while in Argentina, TE is only 56% but $Y$ is on average 98 kg ha$^{-1}$ yr$^{-1}$ [40]. Here, we place TE in Brazil as 81% with $Y$ around 89 kg ha$^{-1}$ yr$^{-1}$.

As expected, the coefficient associated with degraded pasture area shows the lowest effect, in terms of marginal productivity in TE estimation, or $-0.041 \pm 0.006$. The negative sign of the coefficient means that an additional hectare of degraded pasture reduces the average livestock production value. Our results also show that municipalities with larger areas of degraded pasture tend to present lower livestock yields, suggesting that recovery interventions would bring positive gains.

On stocking rates, our results indicate that marginal increases of degraded pasturelands have null or even negative impact on the cattle herd (figure 2). It confirms degraded pasturelands may have no influence on production or even worse, may reduce resources. Null effects could indicate land abandonment; capital outflow; or temporary suspension of activities due to impoverishment of rural areas, as recorded in the poorest regions of Caatinga [41]; long fallow periods in pasture rotation, as carried out in the Pantanal [42]; or land speculation in extensive farms, such as observed in the Pampa [43] and northern Cerrado in Matopiba regions also may explain null effects. Main conclusion is that expanding farms on degraded pastures would be a useless strategy for increasing production [44]. But it can be worse. In some biomes degraded pasture presented negative impact, which may signal persistence of cattle grazing and overgrazing, resulting in a net loss of carrying capacity, commonly registered in the Atlantic Forest [45]; or a 'draining' of cattle ranching due to expansion of the frontier, mainly observed in the Amazon [46]; or progressive pasture replacement to crops, especially in the Cerrado region [44]. In this sense, the more degraded pasture, the lower the production.

On the other hand, planted pasture, in all regions, presents the highest marginal impact on increasing herd size. It generates the highest stocking rate: double or even triple the stocking rate found for native pasturelands. This finding corroborates studies carried out in the field or on experimental farms. Studies have reported that native pastures' carrying capacity ranges from 0.2 to 0.5 heads ha$^{-1}$ due to the low digestible biomass production, low palatability or high biodiversity of herbaceous plants. These qualities force animals to spend more time selecting grasses, thus requiring larger areas for foraging [16].

The stocking rates of planted pastures estimated by the models also presented values very close to those obtained in field studies [16,41–46]. As proposed here, the impact of the recovery of degraded

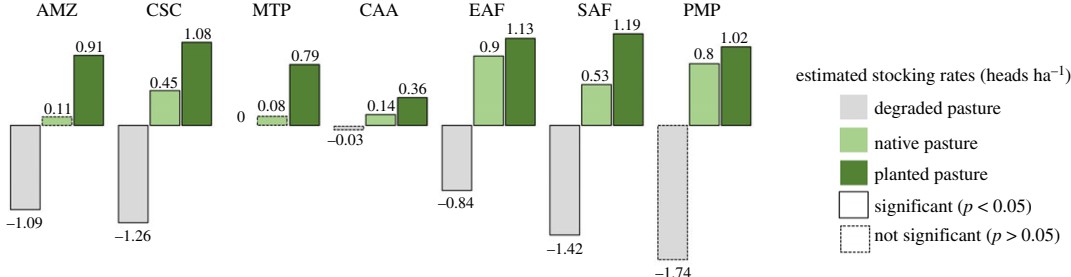

**Figure 2.** Stocking rates estimated for pasture according to regions, using spatial regression models (see electronic supplementary material for details). Stocking rates are measured as additional head per additional hectare of degraded, native and planted pasture. Amazon (AMZ), Matopiba Cerrado (MTP), Caatinga (CAA). Eastern Atlantic Forest (EAF), Pantanal (PTN), Center-Southern Cerrado (CSC), Southern Atlantic Forest (SAF) and Pampa (PMP). PTN region does not have enough observations (municipalities) and was not estimated.

pastures is given by the additional herd expected when all degraded pastures had the same stocking rate of planted pasture, in the respective region. The threshold of each one is determined by the coefficients of regression, instead of biophysical potential, following the positive rather than normative methodological perspective. In this sense, recovery of degraded pasture could increase the cattle herd to 17.7 million bovines in HIS scenario—a net gain equivalent to 9.7% of the current herd. Contributions of the Centre-Southern Cerrado, the Amazon and the Southern Atlantic Forest would be 36%, 31% and 13%, respectively.

Most impressive is that even under the FCC, which prioritizes the allocation of degraded and native pastures to forest restoration, an increase of 9.1 million heads of cattle could be reached, while leaving 12.7 Mha of degraded and native pastures available to cover FC. This would be enough to achieve the Brazilian NDC target (table 1).

Natural areas and restored forests are crucial for the economy and agricultural competitiveness. They supply environmental services such as rainwater irrigation, soil and water conservation, pollinator shelters and climate stability [47,48]. It is imperative to highlight that irrigated systems account for only 3% of all agricultural production value in Brazil, while controlled-environment agriculture such as hydroponic and greenhouse systems share less than 0.5%. This highlights the importance of ecosystem services in Brazilian agriculture.

In addition to this unique contribution of the standing forest—which provides ecosystem services for agriculture as a whole—other conventional land uses, such as raising cattle in native pasturelands, may currently play a greater role in generating direct income and jobs [11]. Therefore, they may hold greater appeal, especially in countries with great social inequality such as Brazil.

Deforestation and pasture degradation have fed back into the process of expanding the agricultural frontier in Brazil [4,12,16]. By contrast, forest restoration and pasture recovery have emerged as two pillars of global food security strategy, transforming degraded areas into the new frontier [49]. Livestock intensification due to pasture recovery would reduce the pressure for new agricultural areas. It could also make room for reforestation, thereby rehabilitating the landscape to provide the environmental services that foster production itself [50]. Recovery of degraded areas for forest restoration and livestock intensification are feasible in Brazil

Of importance is that degraded pastures and deficits are concentrated in specific regions in Brazil. This fact greatly facilitates the adoption of these measures at the regional level (figure 3). Regional concentration will allow focusing public and private resources to leverage pasture recovery. To illustrate this point, consider that a policy to recover 10 Mha of degraded pastures in livestock establishments is defined in ten gradual steps of 1 Mha, by which municipalities are incorporated according to the magnitude of their degraded pasture area. The first million hectares recovered would be concentrated in only 15 municipalities, the second in 27 and the third million in another 41 municipalities. The recovery of 25% of degraded pastures could be achieved by focusing on just 1% of Brazilian municipalities (figure 3).

## 3.2. Financing the recovery of degraded pastures

Our simulations indicate that recovery of degraded pasture can increase the stocking rate, while forest restoration enhances conservation and supports FC compliance. However, several drivers act as

**Table 1.** Degraded, native and planted pasture allocation and additional herd by region in the baseline, HIS and Forest Code compliance (FCC) scenarios.

| | AMZ | CSC | MTP | CAA | EAF | SAF | PMP | total |
|---|---|---|---|---|---|---|---|---|
| **sample** | | | | | | | | |
| municipalities (N) | 401 | 680 | 249 | 937 | 281 | 1962 | 107 | 4617 |
| **baseline** | | | | | | | | |
| degraded pastures (1000 ha) | 2450 | 3202 | 1407 | 2325 | 416 | 966 | 54 | 10 820 |
| native pastures (1000 ha) | 4131 | 8532 | 3259 | 7752 | 1547 | 8139 | 6029 | 39 388 |
| planted pastures (1000 ha) | 31 069 | 30 864 | 6611 | 3115 | 2238 | 15 156 | 3554 | 92 608 |
| FC deficits (1000 ha) | 3746 | 4530 | 878 | 909 | 1079 | 5030 | 695 | 16 866 |
| herd (1000 bovines) | 44 720 | 46 011 | 10 296 | 8647 | 3911 | 40 856 | 7481 | 1 61 921 |
| **HIS scenario** | | | | | | | | |
| degraded pastures recovery (1000 ha) | 2450 | 3202 | 1407 | 2325 | 416 | 966 | 54 | 10 820 |
| additional herd (1000 bovines) | 4914 | 7477 | 1117 | 834 | 819 | 2528 | 54 | 17 744 |
| **FCC scenario** | | | | | | | | |
| degraded pasture to forest restoration (1000 ha) | 1752 | 1684 | 595 | 696 | 319 | 754 | 52 | 5852 |
| degraded pasture recovery (1000 ha) | 698 | 1518 | 812 | 1629 | 98 | 213 | 1 | 4968 |
| native pasture to forest restoration (1000 ha) | 809 | 1689 | 189 | 206 | 514 | 2680 | 638 | 6726 |
| uncovered Forest Code deficit | 1185 | 1157 | 93 | 7 | 246 | 1597 | 4 | 4288 |
| additional herd (1000 bovines) | 3313 | 4901 | 644 | 556 | −5 | 218 | −512 | 9116 |

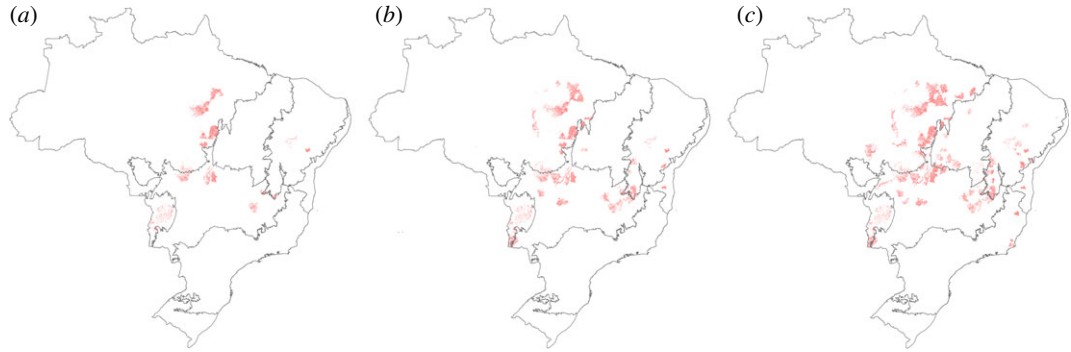

**Figure 3.** Degraded pastures on cattle farms by municipalities (spots). (a) The 15 municipalities with the largest area of degraded pastures hold together 1 Mha of degraded pastures and can potentially add 1.4 million heads, (b) the 42 municipalities that hold together 2 Mha could add 2.79 million heads, and (c) the 83 municipalities that hold together 3 Mha could add 4.02 million heads. Spots have been enlarged for better viewing.

barriers to the recovery of degraded pasture. Most cited are restricted access to rural credit, LT, and insufficient agricultural extension and rural advisory services (RAS) [20,51–53].

We find that efficiency is positively related to RAS ($p < 0.01$): the larger the area of livestock assisted by RAS, the greater the efficiency, in line with other recent studies [52–55]. Brazil once had one of the largest and most efficient public RAS programmes in the world, but they have been in decline since the fiscal crisis in the 1990s. Although they still provide essential services, especially to the poorest producers, they have very limited budgets and human resources, and have even been discontinued in some regions. Private RAS have been consolidating in the country among farmers, usually services involving chemical input and machinery purchases, but they are still little noted among ranchers [54]. In fact, according to the latest Agricultural Census, only 1 out of 10 cattle ranchers declared to have been assisted or to have contracted RAS. More than 77% of livestock farms have no technical support. These figures correspond to 131 million hectares of pastureland left without professional guidance. This chronic problem is not restricted to smallholders in Brazil [11].

Regarding LT, the larger the area of livestock with land title, the greater the efficiency of livestock raising ($p < 0.01$). Despite Brazil having made important advances in the creation of territorial planning instruments, such as the Rural Environmental Registry (CAR), the urgency of better territorial governance in Brazil has been pointed out as a crucial condition to curb deforestation, attracting investments, reducing risks and improving yield [56–58]. Here, we show that LT is also positively related to efficiency.

According to the classical perspective, clear definitions of property rights increase economic efficiency by promoting a fairer distribution of the costs and benefits of production and its externalities, thus encouraging investment [59,60]. However, Ostrom [61] argues that these positive results do not necessarily derive from the institution of private property, but rather from the social recognition of the right to access, use, enjoy and manage resources, whether by the individual or collectively. This is notoriously true in the Amazon [62–64].

LT is also linked to a practical element of financing: access to credit ($p < 0.01$). For the purposes of rural credit, Brazil's official system recognizes both private property and 'right to use' titles, depending upon the legal liability of the public body issuing the land title. In both cases, land title is necessary for the loan guarantee. Even in the case of personal loans, land is generally the asset with the highest value, highest liquidity or lowest risk of depreciation, thus reinforcing land title as the main loan security requirement.

Nevertheless, poor investment capacity is clearly observed throughout Brazil. Analysing aggregated data available from the Brazilian Central Bank (BCB) [65], we identify that between January 2013 and April 2021, a total of US\$ 5.96 billion in rural credit was acquired by ranchers for pasture and soil management. This credit included both investments and working capital, covering a total area of 39 Mha. It represents an average flow of credit inputs of around US\$ 150 ha$^{-1}$ yr$^{-1}$, which is six times lower than the investment levels recommended by the Brazilian Agricultural Research Corporation (Embrapa). We also evaluate disaggregated data on 37 832 rural credit contracts for pastures from 2002 to 2016 (obtained through direct request to BCB, see electronic supplementary material [66]) and find that credit contracts had an average of US\$ 919 ± 37 ha$^{-1}$ yr$^{-1}$. This amount is compatible with

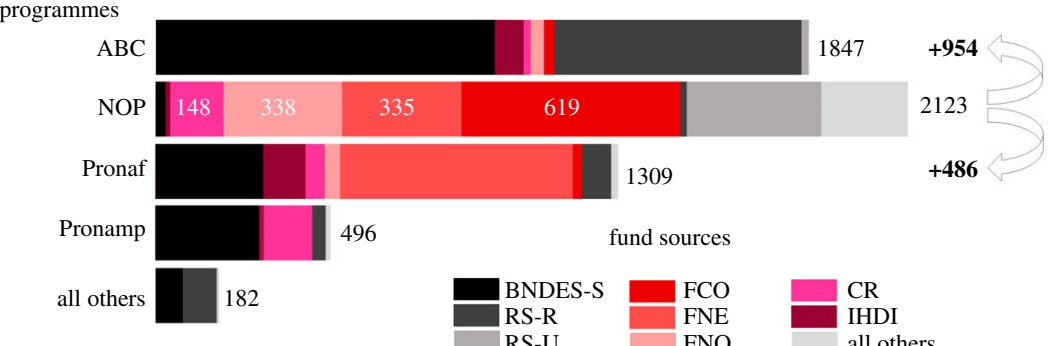

**Figure 4.** Credit programmes and their respective fund sources applied for pasture recovery in Brazil (pasture, tillage, soil fertilization, soil correction and soil protection in livestock activities). Values are total from January 2013 and April 2021, deflated by the National Consumer Price Index (INPC), in US$ million. Programmes are: Low Carbon Agriculture (ABC), National Program for Family Farming (Pronaf), National Program for Medium Sized Producers (Pronamp). Single credits not linked with specific programmes (NOP) and nine other smaller programmes together (all others). Fund sources are: Brazilian Development Bank/Finame with subsidized credit (BNDES-S), Rural Savings with restricted interest rates (RS-R), Rural Savings with unrestricted interest rates (RS-U), Center Western Constitutional Fund (FCO), Northeastern Constitutional Fund (FNE), Northern Constitutional Fund (FNO), Compulsory Resources from checking deposits reserves (CR), Hybrid Capital Debit Instrument (IHDI) and all other sources as unrestricted resources (All others).

Embrapa recommendations, but it covered an average pasture area of $3.93 \pm 1.88$ Mha yr$^{-1}$, representing less than 3% of all pasturelands in the country.

The low level of rural credit acquired by ranchers compared with farmers (agriculture occupies only 29% of total land use, but obtains 71% of all rural credit contracts) may be attributed to several well-documented barriers [20,51–55]. The demand side accounts for several of these barriers: (i) less willingness to hold bank debt; (ii) ranching activity's higher liquidity, which allows anticipating revenues and avoiding loan interest rates; (iii) lower levels of technical and financial planning, and less adherence to RAS services, which both imply less ability to compete for loan resources; and (iv) a higher share of untitled lands. Other challenges arise from the supply end: (i) difficulty tracking the application of resources, especially on breeding farms, (ii) lower levels of loan guarantees linked to land title and tenure, and (iii) fewer sources of funds for livestock.

As noted by Assunção *et al.* [55], there also exists a structural constraint: despite general definitions of rural credit provided by the Safra Plan—the national agriculture and farming financing support plan—the country has dozens of different sources, programmes and lines of rural credit, which compromises efficiency in granting resources. Financing comes from multiple funding sources: either banks or external sources, such as the Brazilian Development Bank (BNDES) or National Treasury funds. Distribution channels are also decentralized through a complex network of bank branches and cooperatives, which are not exclusively dedicated to rural credit. In practice, channels work as retail markets in different programmes, with different credit limits and interest rates. Loan amounts may be determined by banks' own local priorities, spreads and risks rather than the demand for rural credit, agricultural potential or producers' needs [55].

In fact, we identify 11 different sources, and 13 different credit programmes offered for pasture and soil management for ranchers since 2013. Despite so many programmes, most of the loans are still paid in single contracts and not linked to specific programme (NOP), which implies even more dispersion. The most important programme is the Low Carbon Agriculture Program (ABC) which had financed US$ 1.85 billion over the period. The ABC is the financing programme launched in 2011, derived from the National Plan for Low Carbon Emission in Agriculture, led by the Ministry of Agriculture, Livestock and Supply in response to Brazil's commitments affirmed at COP-15 (figure 4).

We test TE specifically for the ABC, and find that the 1600 municipalities in which ABC was responsible for at least 20% of the total credit for pastures showed a higher level of TE, regardless of the region or the provision of agricultural extension and advisory services ($p < 0.01$). One hypothesis for this higher efficiency is that, among all 13 lines of credit, the ABC is the only one that explicitly conditions credit concession on the adoption of pasture recovery practices. Another hypothesis is that ABC has primarily financed regions which already adopt recovery pasture practices [67].

Financial resources from the ABC are not well focused, and have failed to reach priority municipalities. The 11 municipalities which concentrate 10% of degraded pastures in Brazil account

for only 3.5% of total ABC credit disbursements. The 152 municipalities that concentrate 50% of degraded pastures access approximately 20% of ABC's resources. ABC loans may be directed to high-priority municipalities, where pasture recovery can lead to a higher marginal gain, and, therefore, a greater impact on production, as also concluded by Gianetti & Ferreira Filho [67].

The Brazilian Government had just launched the ABC+, the ABC Plan for 2020–2030, and this is the opportunity to make it more efficient, aggregating and consolidating the loans and financing resources as well as redirecting them to priority areas. Since 2013, a total of US$ 954 million of loans applied to pasture recovery came from constitutional funds sources (FCO and FNO), the same ones that finance ABC and other US$ 176 million from rural savings and compulsory resources with restricted and controlled interest rates. If these resources were redirected to the ABC, it would mean an increase of no less than 25% of the resources applied to the recovery of pastures.

Following the same argument of keeping interests rates and spreads, a total of US$ 338 million of constitutional funds for Northeastern region (FNE) made available to NOP could be redirected to the Smallholders Strengthening Program (Pronaf), which would increase the resources for pasture recovery aimed exclusively at poorest family producers and small ranchers by 26%.

Instruments for redirecting or prioritizing rural credit have already proven effective in Brazil. Resolution 3545, issued by the BCB in 2008, conditioned credit in the Amazon on compliance with LT and the FC. Over the last 12 years, this region achieved significantly lower deforestation and higher productivity than its neighbours [53]. Further, society, and agribusiness itself, increasingly demand that the rationale for granting credit must be based on objective criteria that favour good practices. Providing public resources or directing of private resources to agriculture via rural credit needs to be justified by the private and social benefits that result, including conservation of natural resources, common goods and diffuse rights. Clearly, credit directed to pasture recovery meets these criteria, since it leads to a rise in productivity and reduced pressure to exploit new lands.

However, rural credit is not the only and not always the main source of financing for livestock. Since 1970, the share of credit available from the National Rural Credit System—recently through the Safra Plan—for total livestock expenses has varied from 22 to 47% [68–73]. Based on the last Agriculture Census, we estimate that in 2017, around US$ 18 billion in both working capital and investments was expended on livestock in addition to rural credit, mostly included prepaid expenses, borrowing operations outside of rural credit, or private capital [54].

If rural credit should encourage ranchers to recover pasturelands, recognizing the value of externalities and justifying subsidies, then private capital without subsidies can encourage good commercial practices that value producers and their products over the value chain. Enforcing compliance with legislation, and valuing good practices, for example, can be an effective way to finance changes downstream in the production process. It can promote the capitalization of a good producer and increase its competitiveness. Currently, no rural credit line requires or conditions loan concession on traceability.

Here, it is worth highlighting the initiatives of the three largest meat-producing companies in Brazil (which are the first, ninth and 70th in the world). Last year, BRFoods launched its sustainability plan, committing to zero deforestation by 2025 in the grain production chain in the Amazon and Cerrado. Its investment in traceability will be around US$ 34 million by 2025 [74]. In the same year, Marfrig launched the Marfrig Green + Program, with the goal of eliminating deforestation in its supply chain by 2025 for suppliers in the Amazon, and by 2030 for suppliers in the Cerrado. It announced investments of US$ 140 million [75]. Last month, JBS, the largest animal protein company in the world, launched its global programme Net Zero 2040. With investments of US$ 280 billion by 2030, it committed to eliminate direct and indirect carbon emissions in its entire supply chain in the Amazon and Cerrado [76].

These three companies paid over US$ 40 billion to rural producers in 2020. They prompt downstream initiatives in their supply chain, by excluding producers that do not comply with environmental and labour commitments. On the other hand, only 17% of the total US$ 110 billion in net revenues earned by these companies in 2020 came from the domestic market. This shows their strong orientation towards foreign trade. It justifies the effort in quantifying and documenting the impact of domestic demand, to understand how initiatives such as these should be coupled with upstream financing (credit, for example) and traceability mechanisms.

In fact, the Brazilian Coalition on Climate, Forest and Agriculture recently identified two main approaches that may be carried out in parallel to boost private investments while reducing economic and reputational risks: (i) integrate the supply chain for beef, based on jurisdictional models, including monitoring of environmental, fiscal, food safety inspection and labour practices, and (ii) spread the best practices of vertical integration initiatives over the entire chain, from ranchers to retail markets [77].

We propose that degraded pasture recovery should be the main action 'on the ground'. It is truly a new frontier, not only because it represents a great savings of land that has become immobilized in the productive system, but also because it can leverage one of the great comparative advantages: grass-fed systems. Compared with American and European grain-fed or feedlot systems, the Brazilian grass-fed grazing system has many advantages that are valued in the international market. These include (i) better sanitary conditions, and less risk of contamination by pathogens or development of animal or human diseases; (ii) less need for medicines; (iii) greater animal comfort; and (iv) less competition for access to chemical inputs and raw materials, including grains that could be directly consumed by humans [78].

# 4. Conclusion

Inefficiency of Brazilian livestock production is very high, around 18%. This is positively correlated to degraded pastures. Recovery of degraded pastures, starting with 12 Mha already recognized by ranchers, is an opportunity to increase the productivity and competitiveness of Brazil's livestock sector in the short term. Additionally, it would help the sector to relinquish its stigma of being the world's largest driver of deforestation.

Brazil's degraded pastures are highly clustered in a few regions. This makes them easier targets for recovery programmes, ensuring greater effectiveness in converting these pastures' productivity. We observe that 25% of Brazil's degraded pastures are clustered in just 1% of its municipalities.

Rural credit can have a significant impact in reducing inefficiency. This is an important bottleneck for economic and productive gains. On average, livestock farms invest 7–30 times less than necessary to recover pastures. On the other hand, rural credit finances only US$ 1 of every US$ 4 invested in livestock. Therefore, it is important to redirect working capital for investment.

The ABC Program, especially the subprogramme 'Recovery of Degraded Pastures', must be broadly expanded. One first step can be redirecting resources from rural savings and constitutional funds with controlled interest, and from currently available funds, to promote pasture recovery, without being linked to specific programmes.

Data accessibility. Data are available from the Dryad Digital Repository: https://doi.org/10.5061/dryad.ksn02v740 [79]. Electronic supplementary material available [80].

Authors' contributions. R.F.-B. carried out literature review, database collection, developed spatial regression model scripts and analysis, led the design of the study and led the manuscript. J.G.F. contributed to literature review, database collection, led the development of the stochastic frontier model, participated in the design of the study and contributed to the manuscript.

Competing interests. We declare we have no competing interests.

Funding. This research was originally funded by World Resources Institute, New Climate Economy, Good Energies, Moore Foundation and by the National Institute of Science and Technology for Climate Change-Phase 2 project (INCT MC Phase 2): FAPESP grant no. 2014/50848-9, 964, National Coordination for High Level Education and Training (CAPES) grant no. 88887.136402-00 and CNPq grant no. 465501/2014-1.

Acknowledgements. The authors express their deep gratitude for the valuable discussions and voluntary contributions received from Joaquim Levy, João Adrien, Leia Harfuch, Leonardo Fleck and Mariane Crespolini dos Santos. The authors are also grateful to Berta Pinheiro, Carolina Genin, Helen Mountford, Marcelo Matsumoto, Miguel Calmon, Leonardo Barbosa, Paulo Camuri, Rachel Biderman, Talita Esturba and Viviane Romeiro for their voluntary scientific and technical support. We are also very grateful to the anonymous referee for suggestions and constructive criticism.

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
