## [Peer Review File · Royal Society Open Science]

Review History

RSOS-201854.R0 (Original submission)

Review form: Reviewer 1

Is the manuscript scientifically sound in its present form?

Yes

Are the interpretations and conclusions justified by the results?

Yes

Is the language acceptable?

Yes

Do you have any ethical concerns with this paper?

Yes

Have you any concerns about statistical analyses in this paper?

No

Recommendation?

Accept with minor revision (please list in comments)

Comments to the Author(s)

An important topic in the context of the Brazilian NDC and its targets related to land use. Appropriate timing. The paper needs a few improvements and a more detailed list can be found in the attached file (see Appendix A).

Review form: Reviewer 2**Is the manuscript scientifically sound in its present form?**

No

Are the interpretations and conclusions justified by the results?

Yes

Is the language acceptable?

No

Is it clear how to access all supporting data?

No

Do you have any ethical concerns with this paper?

No

Recommendation?

Major revision is needed (please make suggestions in comments)

Comments to the Author(s)

The authors make the important point that pasture productivity could be improved in Brazil, which would hopefully decrease agricultural pressure on intact forest. I agree that this is an important message but the manuscript will need substantial revisions (detailed below) to be suitable for publication in Royal Society Open Science and accessible to a broad audience. I also made detailed edits on the manuscript (see Appendix B).

1. The authors never explain early on why improving pasture productivity entails and this is important to understanding later portions of the manuscript. I think they mean fertilizing the pastures, moving cattle more frequently, and better animal management generally, but the specific practices need to be clearly described. A related important point is that the authors seem to use pasture restoration and recovery interchangeably, which is misleading. It seems that they are talking about improving cattle productivity (the term I suggest they use), rather than improving habitat quality in pastures. There is an extensive literature on silvopastoral systems (e.g., Murgueitio et al., 2011; Calle et al., 2013; Calle, 2020), which aims to do both, and I strongly recommend that the authors consider this literature and what might be achievable in Brazil. Regardless the authors need to clarify their use of terminology. What they are describing is not "restoration" (Gann et al., 2019).
2. The writing was hard to follow at multiple places in the article for a number of reasons. The authors repeat a few generalities many times (e.g. increasing pasture productivity is the new frontier), but then jam multiple technical terms into a single sentence without much background for the general reader. I have noted many cases of this in the paper. The authors alternate between some short paragraphs (just one sentence) and then very long sentences with multiple

clauses. I note several that I found unclear. More generally, there are many run on sentences, missing words, and English grammar errors that made the paper hard to follow. Once the other revisions are completed, the paper should be reviewed by a native English speaker.

3. The authors refer almost exclusively to Brazil and references from Brazil in this paper. I realize that Brazil is a large and complex country, but for an international journal the authors should consider how the Brazilian experience compares to other systems and countries. For example, there is an extensive literature on sustainable cattle ranching in Colombia. The authors also assume that various laws and organizations in Brazil (e.g. Embrapa, Safra Plan, ABC) are known to readers. Certainly, some readers of this article will know those terms, but for a general journal this knowledge should not be assumed and acronyms should always be spelled out. The authors assume a lot of knowledge about Brazilian agricultural credit.

4. As noted below, multiple figures are lacking sufficiently descriptive captions.

A few more specific comments in addition to the many directly on the manuscript.

Title. It's not clear to the reader what the degraded pastures in Brazil are the new frontier for.

They could be the new frontier for soybean growth given that the title is vague. The title needs to better communicate the main message of the paper.

Figure 1. A much more detailed caption is needed so the figure can be read and understood separately from the paper. This is true of several figures (e.g. Fig. 3 and 6 also).

Figure 3. The line for the y-axis is missing and a more descriptive caption is needed.

Figure 4. Is too small to be legible.

Literature cited

Calle, A. (2020) Partnering with cattle ranchers for forest landscape restoration. *Ambio*, 49, 593-604. [10.1007/s13280-019-01224-8](https://doi.org/10.1007/s13280-019-01224-8)

Calle, Z., Murgueitio, E., Chará, J., Molina, C.H., Zuluaga, A.F. & Calle, A. (2013) A strategy for scaling-up intensive silvopastoral systems in Colombia. *Journal of Sustainable Forestry*, 32, 677-693.

Gann, G.D., McDonald, T., Walder, B., Aronson, J., Nelson, C.R., Jonson, J., . . . Dixon, K.W. (2019) International principles and standards for the practice of ecological restoration. Second edition. *Restoration Ecology*, 27, S1-S46. [10.1111/rec.13035](https://doi.org/10.1111/rec.13035)

Murgueitio, E., Calle, Z., Uribe, F., Calle, A. & Solorio, B. (2011) Native trees and shrubs for the productive rehabilitation of tropical cattle ranching lands. *Forest Ecology and Management*, 261, 1654-1663. [10.1016/j.foreco.2010.09.027](https://doi.org/10.1016/j.foreco.2010.09.027)

Decision letter (RSOS-201854.R0)

Dear Dr Feltran-Barbieri

The Editors assigned to your paper RSOS-201854 "Degraded Pastures in Brazil: the new frontier" have now received comments from reviewers and would like you to revise the paper in accordance with the reviewer comments and any comments from the Editors. Please note this decision does not guarantee eventual acceptance.

Please submit your revised manuscript and required files (see below) no later than 21 days from today's (8th of March 2021). Note: the ScholarOne system will 'lock' if submission of the revision is attempted 21 or more days after the deadline. If you do not think you will be able to meet this deadline please contact the editorial office immediately.

on behalf of Dr Agnieszka Latawiec (Subject Editor)
openscience@royalsociety.org

Associate Editor Comments to Author (Dr Agnieszka Latawiec):

Dear Authors

Please incorporate the suggestions of both reviewers, with a special attention to the comments of the second reviewer.

Kind Regards
Agnieszka Latawiec

Reviewer comments to Author:

Reviewer: 1

Comments to the Author(s)

An important topic in the context of the Brazilian NDC and its targets related to land use. Appropriate timing. The paper needs a few improvements and a more detailed list can be found in the attached file.

Reviewer: 2

Comments to the Author(s)

The authors make the important point that pasture productivity could be improved in Brazil, which would hopefully decrease agricultural pressure on intact forest. I agree that this is an important message but the manuscript will need substantial revisions (detailed below) to be suitable for publication in Royal Society Open Science and accessible to a broad audience. I also made detailed edits on the manuscript.

1. The authors never explain early on what improving pasture productivity entails and this is important to understanding later portions of the manuscript. I think they mean fertilizing the pastures, moving cattle more frequently, and better animal management generally, but the specific practices need to be clearly described. A related important point is that the authors seem to use pasture restoration and recovery interchangeably, which is misleading. It seems that they are talking about improving cattle productivity (the term I suggest they use), rather than improving habitat quality in pastures. There is an extensive literature on silvopastoral systems (e.g., Murgueitio et al., 2011; Calle et al., 2013; Calle, 2020), which aims to do both, and I strongly recommend that the authors consider this literature and what might be achievable in Brazil. Regardless the authors need to clarify their use of terminology. What they are describing is not “restoration” (Gann et al., 2019).

2. The writing was hard to follow at multiple places in the article for a number of reasons. The authors repeat a few generalities many times (e.g. increasing pasture productivity is the new frontier), but then jam multiple technical terms into a single sentence without much background for the general reader. I have noted many cases of this in the paper. The authors alternate between some short paragraphs (just one sentence) and then very long sentences with multiple clauses. I note several that I found unclear. More generally, there are many run on sentences, missing words, and English grammar errors that made the paper hard to follow. Once the other revisions are completed, the paper should be reviewed by a native English speaker.

3. The authors refer almost exclusively to Brazil and references from Brazil in this paper. I realize that Brazil is a large and complex country, but for an international journal the authors should consider how the Brazilian experience compares to other systems and countries. For example, there is an extensive literature on sustainable cattle ranching in Colombia. The authors also assume that various laws and organizations in Brazil (e.g. Embrapa, Safra Plan, ABC) are known to readers. Certainly, some readers of this article will know those terms, but for a general journal this knowledge should not be assumed and acronyms should always be spelled out. The authors assume a lot of knowledge about Brazilian agricultural credit.

4. As noted below, multiple figures are lacking sufficiently descriptive captions.

A few more specific comments in addition to the many directly on the manuscript.

Title. It's not clear to the reader what the degraded pastures in Brazil are the new frontier for.

They could be the new frontier for soybean growth given that the title is vague. The title needs to better communicate the main message of the paper.

Figure 1. A much more detailed caption is needed so the figure can be read and understood separately from the paper. This is true of several figures (e.g. Fig. 3 and 6 also).

Figure 3. The line for the y-axis is missing and a more descriptive caption is needed.

Figure 4. Is too small to be legible.

Literature cited

Calle, A. (2020) Partnering with cattle ranchers for forest landscape restoration. *Ambio*, 49, 593-604. [10.1007/s13280-019-01224-8](https://doi.org/10.1007/s13280-019-01224-8)

Calle, Z., Murgueitio, E., Chará, J., Molina, C.H., Zuluaga, A.F. & Calle, A. (2013) A strategy for scaling-up intensive silvopastoral systems in Colombia. *Journal of Sustainable Forestry*, 32, 677-693.

Gann, G.D., McDonald, T., Walder, B., Aronson, J., Nelson, C.R., Jonson, J., . . . Dixon, K.W. (2019) International principles and standards for the practice of ecological restoration. Second edition. *Restoration Ecology*, 27, S1-S46. [10.1111/rec.13035](https://doi.org/10.1111/rec.13035)

Murgueitio, E., Calle, Z., Uribe, F., Calle, A. & Solorio, B. (2011) Native trees and shrubs for the productive rehabilitation of tropical cattle ranching lands. *Forest Ecology and Management*, 261, 1654-1663. [10.1016/j.foreco.2010.09.027](https://doi.org/10.1016/j.foreco.2010.09.027)

===PREPARING YOUR MANUSCRIPT===

Your revised paper should include the changes requested by the referees and Editors of your manuscript. You should provide two versions of this manuscript and both versions must be provided in an editable format:
 one version identifying all the changes that have been made (for instance, in coloured highlight, in bold text, or tracked changes);
 a 'clean' version of the new manuscript that incorporates the changes made, but does not highlight them. This version will be used for typesetting if your manuscript is accepted.
 Please ensure that any equations included in the paper are editable text and not embedded images.

===PREPARING YOUR REVISION IN SCHOLARONE===

- An editable file of each table (.doc, .docx, .xls, .xlsx, or .csv).
 - An editable file of all figure and table captions.
- Note: you may upload the figure, table, and caption files in a single Zip folder.
- Any electronic supplementary material (ESM).
 - If you are requesting a discretionary waiver for the article processing charge, the waiver form must be included at this step.
 - If you are providing image files for potential cover images, please upload these at this step, and inform the editorial office you have done so. You must hold the copyright to any image provided.
 - A copy of your point-by-point response to referees and Editors. This will expedite the preparation of your proof.

- Ensure that your data access statement meets the requirements at <https://royalsociety.org/journals/authors/author-guidelines/#data>. You should ensure that you cite the dataset in your reference list. If you have deposited data etc in the Dryad repository, please include both the 'For publication' link and 'For review' link at this stage.
- If you are requesting an article processing charge waiver, you must select the relevant waiver option (if requesting a discretionary waiver, the form should have been uploaded at Step 3 'File upload' above).
- If you have uploaded ESM files, please ensure you follow the guidance at <https://royalsociety.org/journals/authors/author-guidelines/#supplementary-material> to include a suitable title and informative caption. An example of appropriate titling and captioning may be found at https://figshare.com/articles/Table_S2_from_Is_there_a_trade-off_between_peak_performance_and_performance_breadth_across_temperatures_for_aerobic_scope_in_teleost_fishes_/3843624.

Author's Response to Decision Letter for (RSOS-201854.R0)

See Appendix C.

RSOS-201854.R1 (Revision)

Review form: Reviewer 1

Is the manuscript scientifically sound in its present form?

Yes

Are the interpretations and conclusions justified by the results?

Yes

Is the language acceptable?

Yes

Do you have any ethical concerns with this paper?

Yes

Have you any concerns about statistical analyses in this paper?

No

Recommendation?

Accept with minor revision (please list in comments)

Comments to the Author(s)

Minor specific comments.

Pg 6/52 line 51 - indicate where we can find the "controllers" variables in SM.

Pg 10/52 lines 40-52 - it looks that a call to figure 2 is missing, as authors are talking about it.

Pg 10/52 lines 54-60 - indicate where to find the results presented. Also, in this paragraph, the authors start by presenting planted pasture but present an argument about natural pasture to corroborate the initial statement, which seems inconsistent or at least not clear.

Pg 11/52 lines 31-37 - suggest authors indicate Table 1 here.

Pg 11/52 lines 39 - FCS or FCC as indicated in Table 1?

Pg 12/52 lines 41-54 - The message is not clear and it is floating here with poor connection with the previous paragraph.

Pg 15/52 lines 40-44 - If I understood correctly, the authors mention the volume of credit available, but I missed how much was effectively used to finance livestock activities.

Pg 17/52 line 20 - A typo "meet" when the correct would be "meat".

SM - Table SM1 - variable "degpast" description seems to be incorrect.

Review form: Reviewer 2

Is the manuscript scientifically sound in its present form?

Yes

Are the interpretations and conclusions justified by the results?

Yes

Is the language acceptable?

No

Do you have any ethical concerns with this paper?

No

Have you any concerns about statistical analyses in this paper?

No

Recommendation?

Major revision is needed (please make suggestions in comments)

Comments to the Author(s)

As I noted in the last version, I think the authors make an important point that pasture productivity could be improved in Brazil, which would hopefully decrease agricultural pressure on intact forest. I appreciate that the authors did a major reorganization of the paper so that there are fewer short paragraphs and the text flows better. They also expanded their description of different approaches to improve pasture productivity.

There are however, some issues that still need to be resolved.

1. The paper could make the same point in a lot less words. It's a long article and could easily be 20% shorter without losing any key points.
2. The paper needs a careful editing. There are several grammatical errors, typos, or incorrect word choices on each page. I did a more thorough editing last time but that was beyond the responsibility of a reviewer.
3. The title could be more informative. It says that degraded pastures are the new frontier but the reader doesn't know what they are the frontier for. I think a more accurate title would be "Improving pasture productivity in Brazil can have major economic and ecological benefits".
4. As I noted in the last version, what the authors describe is not "restoration" per se. "Restoring" those habitats would mean removing them from pasture land and restoring native ecosystem types. In the revised version, they use the word "restore" and "restoration" more in the abstract than the last version. The terminology they use in the text of the paper such as "improving pasture productivity", "restoring pasture productivity", or "recovering pastures" is fine. But they need to fix the terminology in the abstract where they use the term "restor*" incorrectly multiple times.

A few of the many minor edits

p. 3 line 12 – Should be "a" depreciation.

p. 3 line 26 "would permit to rip off" needs to be rewritten. Maybe "would allow".

p. 3 line 28 in "the" Paris.

p. 3 line 49 "has performed"

p. 3 line 60 "recur" does not make sense here.

There were so many grammatical errors that I stopped correcting them after p. 3.

p. 9 line 52. There is no need to repeat the sentence about TE in Brazil being 81% since that number is in the prior paragraph.

p. 9 final para. This entire paragraph says that if productivity of pasture is low then it can be improved. That goes without saying. The paragraph could be deleted.

p. 11 line 31. Estimated as the "difference" between...

p. 11 line 40. Should be 9.1 M heads... Need to carefully review throughout that decimal points are used before 10ths digits, not commas (which I realize is what is done on in most Latin American countries).

p. 14 line 19 "Elinor" Ostrom.

Decision letter (RSOS-201854.R1)

Dear Dr Feltran-Barbieri,

It is a pleasure to accept your manuscript entitled "Degraded Pastures in Brazil: the new frontier" in its current form for publication in Royal Society Open Science. The comments of the reviewer(s) who reviewed your manuscript are included at the foot of this letter.

You can expect to receive a proof of your article in the near future. Please contact the editorial office (openscience@royalsociety.org) and the production office (openscience_proofs@royalsociety.org) to let us know if you are likely to be away from e-mail contact – if you are going to be away, please nominate a co-author (if available) to manage the proofing process, and ensure they are copied into your email to the journal.

on behalf of Dr Agnieszka Latawiec (Associate Editor) and Agnieszka Latawiec (Subject Editor)
openscience@royalsociety.org

Reviewer comments to Author:
Reviewer: 1
Comments to the Author(s)
Minor specific comments.

Pg 6/52 line 51 – indicate where we can find the “controllers” variables in SM.

Pg 10/52 lines 40-52 – it looks that a call to figure 2 is missing, as authors are talking about it.

Pg 10/52 lines 54-60 – indicate where to find the results presented. Also, in this paragraph, the authors start by presenting planted pasture but present an argument about natural pasture to corroborate the initial statement, which seems inconsistent or at least not clear.

Pg 11/52 lines 31-37 – suggest authors indicate Table 1 here.

Pg 11/52 lines 39 – FCS or FCC as indicated in Table 1?

Pg 12/52 lines 41-54 – The message is not clear and it is floating here with poor connection with the previous paragraph.

Pg 15/52 lines 40-44 – If I understood correctly, the authors mention the volume of credit available, but I missed how much was effectively used to finance livestock activities.

Pg 17/52 line 20 – A typo “meet” when the correct would be “meat”.

SM – Table SM1 – variable “degpast” description seems to be incorrect.

Reviewer: 2

Comments to the Author(s)

As I noted in the last version, I think the authors make an important point that pasture productivity could be improved in Brazil, which would hopefully decrease agricultural pressure on intact forest. I appreciate that the authors did a major reorganization of the paper so that there are fewer short paragraphs and the text flows better. They also expanded their description of different approaches to improve pasture productivity.

There are however, some issues that still need to be resolved.

1. The paper could make the same point in a lot less words. It's a long article and could easily be 20% shorter without losing any key points.
2. The paper needs a careful editing. There are several grammatical errors, typos, or incorrect word choices on each page. I did a more thorough editing last time but that was beyond the responsibility of a reviewer.
3. The title could be more informative. It says that degraded pastures are the new frontier but the reader doesn't know what they are the frontier for. I think a more accurate title would be “Improving pasture productivity in Brazil can have major economic and ecological benefits”.
4. As I noted in the last version, what the authors describe is not “restoration” per se. “Restoring” those habitats would mean removing them from pasture land and restoring” native ecosystem types. In the revised version, they use the word “restore” and “restoration” more in the abstract than the last version. The terminology they use in the text of the paper such as “improving pasture productivity”, “restoring pasture productivity”, or “recovering pastures” is fine. But they need to fix the terminology in the abstract where they use the term “restor*” incorrectly multiple times.

A few of the many minor edits

p. 3 line 12 – Should be “a” depreciation.

p. 3 line 26 “would permit to rip off” needs to be rewritten. Maybe “would allow”.

p. 3 line 28 in “the” Paris.

p. 3 line 49 “has performed”

p. 3 line 60 “recur” does not make sense here.

There were so many grammatical errors that I stopped correcting them after p. 3.

p. 9 line 52. There is no need to repeat the sentence about TE in Brazil being 81% since that number is in the prior paragraph.

p. 9 final para. This entire paragraph says that if productivity of pasture is low then it can be improved. That goes without saying. The paragraph could be deleted.

p. 11 line 31. Estimated as the “difference” between...

p. 11 line 40. Should be 9.1 M heads... Need to carefully review throughout that decimal points are used before 10ths digits, not commas (which I realize is what is done on in most Latin American countries).

p. 14 line 19 “Elinor” Ostrom.

Royal Society Open Science peer-review - RSOS-201854**Summary**

The authors elaborate very well an important theme for changing the course of GHG emissions in Brazil, focused on land use in particular, in pastures. Showing with a robust analysis and consistent data from the Brazilian government, the importance of degraded and underutilized areas that should be part of the priorities of government policies and actions at different levels, federal, state and municipal, in the indicated areas that concentrate most of these pastures degraded.

General comments

1. Clear objectives, Well designed and sound analysis with Spatial component.
2. Very important topic connected to the main problem of Brazilian GHG emissions, Land use (almost 70% of Brazilian emissions), pastures and cows contribute significantly to it.
3. In Material and Methods, to be clearer for the reader, could explicitly indicate that variables and more detailed information can be found in SM, for instance a map with the 8 regions and the list of municipalities included in each one, and the variables used for the models.
4. There is room for English improvements and typo review (there are some Portuguese words like in page 12 line 28.)
5. It is recommended to change the currency to US\$, as this is a global Journal, and readers would understand better the values in a global currency.
6. The figures do not have a clear indication throughout the text, to simplify the reader the time to consult them. The same apply for the Supplement Material.
7. For the benefit of readers, it is important to clarify what ABC program is in the supplement material, in order to highlight the importance of this program in view of the needs of a low carbon economy.
8. As livestock has a major contribution to Brazilian emissions, it would be advisable to include information on the gains from the results of this work and the indicative measures proposed. Thus, the work can gain a good alignment with low carbon economy discussion.

Specific comments

Page 3 lines 16-21: the complementarity message is not very clear, as it explains only the extensive side of it. Could be clearer.

Page 3 lines 60-62: Would recommend a source for the 100-120M ha of degraded pasture.

Page 4 lines 3-7: Message is not clear, could add information and elaborate a little bit more on degraded pastures, as it could also be associated with abundance of low-price lands in the amazon against the high costs to recover the pasture, and the recovery “opportunity” is not so clear from the rancher perspective.

Page 4 line 52: typo “et al”.

Page 5 line 5: substitute “complexity” with “complex”

Page 7 line 27: where it is: “all 5,563 the municipalities” change to “all the 5,563 municipalities”

Page 7 line 27-28: Would be useful for the reader to have a map showing the 8 regions and the list of municipalities in each one to have a clear picture of the scenario and replicability of the analysis.

Page 7 line 36 and 38: I suggest changing the “Forest Code” for the “Law of Protection of Native Vegetation” (LPVN) it is more appropriate.

Page 8 line 51: The reader would benefit from an indication of the reasons why “an effort of this magnitude is doubtful to be possible”, recommend to elaborate in a concise way.

Page 9: the map should be bigger, as the legend is almost impossible to read. In addition, the SM should have the list of municipalities included in each region. This map could be presented in Material and Methods, leaving here only the table.

Page 10 Figure 3: It is not easy to follow the colours for “Amazon” and “Eastern Atlantic Forest”, as it looks like it is not in order, and slightly different in graph and legend.

Page 10 lines 39-46: the message is not clear, could be more specific and clearer about the laws the authors refer to. Everything described is defined in the same legislation or in different ones. It is a good point but need clarity.

Page 12 lines 23-30: How much does this represent of the total credit for the period?

Page 12 lines 33-34: Would be good to have numbers (%) here, demonstrating how much it has grown.

Page 13 lines 24-34: I recommend the authors presenting the size of the ABC program (in dollars) and the portion of it destined to pasture recovery, so that it is clear the amount contracted in the priority municipalities for pasture recovery, including its size in relation to traditional credits, to showcase that a program important as this to take the necessary path towards a low carbon economy / agriculture, it is still well below the volume of traditional credit.

Page 13 lines 37-50: The message is clear to move budget to ABC, but what do the authors recommend to make this really happen?

Page 14 Figure 6: I assume the amount represented is related to pasture recovery, if so, what is the size of these programs and their main objectives (suggestion: a table or short text in Supplement Material about the programs).

Page 14 lines 44-50: I suggest including a concrete example to reinforce the point.

Page 15 line 28: Message is not clear as a final statement of results and discussion.

Supplement Material

SM 1.1 Second paragraph: “The census did not provide microdata - producer-level responses.” Would recommend to clarify the reason for not having property level data. It is related to legislation restrictions like the Centra Bank data?

SM 1.3 table: degpast and plantpast have the same description, is it correct?

SM 1.3 table: labor, aterarea and aterper have 5,662 and 5,663 municipalities, is it correct? According to IBGE, Brazil has 5,570 municipalities.

SM 2.1: please indicate the plataform/program and version used to run the analysis. (Stata?)

Appendix B

R. Soc. open sci. article template

ROYAL SOCIETY
OPEN SCIENCE

R. Soc. open sci.
doi:10.1098/not yet assigned

Degraded Pastures in Brazil: the new frontier

Rafael Feltran-Barbieri*[†] & José Gustavo Féres^{‡,¶}

[†] World Resources Institute Brazil, Rua Claudio Soares 72 sala 1510, Sao Paulo SP, Brazil,05422-030

[‡] Institute for Applied Economic Research (IPEA), Avenida Presidente Vargas 730, 17^o Andar, Rio de Janeiro RJ, Brazil,20071-900

[¶] Brazilian School of Economics and Finance (EPGE-FGV), Praia de Botafogo 190, 11^o Andar, Rio de Janeiro RJ 22250-900

Keywords: Degraded Pasturelands, Livestock Efficiency, Rural Credit, Brazil

Abstract

Degraded pasture restoration is ~~at~~ the major liability in Brazilian agriculture but could be ~~the~~ a main asset. Here we show the technical inefficiency of livestock activity in Brazil is around 19% in which degraded pastures ~~being~~ are the main factor of diseconomy of scale. On the other hand, the recovery of 12 million hectares of degraded pastures could generate an additional production of 16.9 million bovines. There is a large regional concentration of degraded pastures, which would facilitate the targeting of Technical Assistance and Rural Extension (ATER) and Rural Credit efforts to foster pasture recovery, as these two factors have a significant impact in reducing inefficiency. 1% of Brazilian municipalities concentrate 25% of degraded pastures that could generate almost 30% of the total ~~increment~~ increase of the additional herd via pasture recovery. At the municipality level, even if most of the degraded pastures were allocated to forest recovery in order to comply with the Forest Code, an increase of 9 M of cattle would be possible due to the recovery of degraded pastures that would exceed the minimum necessary to comply with the environmental law. Redirecting investment credits, especially for resources with controlled interest rate, ~~Better management of d~~ Degraded pasturelands ~~is~~ are the new frontier that could boost livestock activities and avoid deforestation in Brazil.

Commented [Rev1]: Is it the pasture restoration or the degraded pastures that are the liability?

Commented [Rev2]: Verb missing in this sentence.

1. Introduction

Brazil produces 16% of the world's beef and accounts for 20% of the global beef markets [1], having traded in 2019 about US \$ 7.6 billion FOB (Free on Board) [2], half of which to China and Hong Kong. One third of the Brazilian Agribusiness GDP, or about US \$ 81 billion, and a quarter of the Gross Value of Agriculture, around US \$ 15 billion, ~~are~~ is generated by cattle livestock, which employs 3 million people in rural areas in Brazil [3,4].

*Author for correspondence (rafael.barbieri@wri.org).

[†]Present address: Department, Institution, Address, City, Code, Country

Despite this large-size economy, Brazilian beef has performing far below its biophysical potential, producing an average of 0.87 AU/ha where biocapacity exceeds 2.94 AU/ha [5]. There is normally a great variation between the different regions of the country, but even with enormous regional heterogeneity and a high degree of specialization of production - e.g. breeding, recreating, fattening and semi-confinement - national averages are instructive ~~to as~~ revealing the large share of ~~very~~ extensive and inefficient systems.

Unlike crops as soybeans and sugarcane - in which the incorporation of technology is a condition of competitiveness - in livestock production the production factors are more interchangeable, usually allowing additional allocation of land at the expense of additional capital. It helps to explain the coexistence of ultra-extensive farms in the Amazon and the high-tech semi-confinement systems in the Center-South of Brazil. High liquidity - a cow may always be sold, wherever its age or weight - is a counterpoint to low profitability, and land speculation as a value store activity is an alternative to low return of productive capital, which reinforces the incentive to expand agricultural frontiers [6].

From the rancher rationality to the logic of the supply chain, extensive and intensive livestock systems may be complementary rather than substitutes. As the large slaughterhouse conglomerates increase their purchasing power over the territory, ultra-extensive systems become more integrated, supplying the fattening farms and pushing the formation of new herds that can foster the beef chain.

It is precisely this integration under the perspective of a new market tide which pushes livestock forward. More pasturelands mean more deforestation. In the last 35 years 45 Mha - a 1.8-fold UK area - of new pastures were ~~grown-created~~ in Brazil [7]. During this time, a huge area of 64 Mha ~~where~~ deforested and immediately replaced by new pastures, while 18 M ha of pre-existing pastures have been replaced by agriculture, forestry and dams [7].

Consequently, 70% of the current pastures in the Amazon came from deforestation since 1985 - or 37 M ha, while in the Cerrado and Atlantic Forest, a third of current pastures originated from deforestation in the last 35 years. This vertiginous pastureland expansion at the expenses of deforestation occurred mainly until the early 2000s, slowing down to less than 20% in 2012, stabilizing the area of pastures ~~in at on~~ 180 Mha all over the country [7].

In the last decade carcass weight increased by only 10% (0.74% p.y.) and a slight advance in the stocking rate from 0.85 to 0.87 AU / ha was observed. Meanwhile corn productivity grew by 5.3% p.y. and soybean 3.9% p.y., leaving levels already competitive in relation to the neighbours of Mercosul and even the USA [4].

Deforestation has ~~raised-increased~~ since 2012 and surged up last year by 30%. 3.4 Mha were deforested in the last 7 years, 94% of which ~~was~~ due new pasturelands. [7] 99% of the 1Mha deforested in Amazon last year was illegal [8].

The negative fiscal, legal, environmental and social effects of the frontier expansion due to illegal deforestation have been proved ~~to be~~ inefficient, perverse and unacceptable by the ~~own~~-Brazilian agribusiness sector. Recently, the Brazilian Coalition on Climate, Forest and Agriculture, a multi-sectorial agreement with 250 institutions, including the Brazilian Agribusiness Association (ABAG), Brazilian Beef Exporters Association (ABIEC) and the ~~three~~3 major private Banks - with more than US 700 Bn in assets - ~~have~~s announced a proposal for stop deforestation, including tracing production chain and cutting investments ~~in~~ producers and industries directly or indirectly responsible for illegal deforestation [9].

Commented [Rev3]: There are a lot of two sentence paragraphs in a row that make the writing rather choppy. I suggest combining a couple of these paragraphs around overarching themes.

Commented [Rev4]: Unclear.

Commented [Rev5]: Write out numbers when they start sentences.

These pressures put the livestock growth model based on extensive practices in check, and open a new opportunity in a new frontier: the degraded lands. Brazil has between 100 and 120 Mha of degraded pastures – equivalent to all the EU's pasturelands area.

Low investments, overgrazing or land abandonment as a result of ~~the own~~this extensive grazing system are the main causes of increases of land degradation in Brazil [10,11]. On the other hand, the recovery of degraded pastures is the main strategy that the producer has at ~~his~~their fingertips to start this trajectory, being the first and main beneficiary of the production gain and legal security, from which the benefits can reverberate for the entire chain .

Commented [Rev6]: Could include the two paragraphs on deforestation here to reduce the number of many short paragraphs.

This article seeks ~~to contribute~~ to elucidate the impacts of degraded pastures in Brazilian livestock production, how the recovery of these pastures could leverage the activity, and what mechanisms could be used or improved to trigger this productive revolution. The main objectives are

- (1) Estimate the technical efficiency of livestock in Brazil, and, for the first time, identify marginal impact - the stocking rate - of planted, native and degraded pasturelands in all Biomes,
- (2) Estimate the potential increase ~~of in~~ cattle that would be achieved by recovering degraded pasturelands, and
- (3) Identify priority sites and policies that should be the start points ~~to of~~ a new livestock system.

Differently from any ~~other~~ previous study, which used GIS tools to identify and measure impact of recovery of the degraded pasturelands [12, 13, 14, 15], we focused on producer-perspective, using data from the latest Agricultural Census [4]. The recognition of the of degraded pastures by the ~~own~~Brazilian rural producers is a crucial step to encourage any pasturelands recovery initiative. Further, in contrast with the biophysical potential as usually referenced as the threshold of potential cattle increment, we used the regional stocking rate of non-degraded pasturelands locally estimated by Spatial Error Model Regressions (SEM), applied by biome at the municipality level. This approach provides more realistic threshold and reflects the average ~~of~~ stocking rate already achieved by local ranchers, measured in marginal effects.

2. Material and Methods

Commented [Rev7]: I stopped detailed editing here but the rest of the paper needs to be carefully proofread by a native English speaker.

2.1 Degraded pasturelands

Degraded pasturelands are native or planted pastures with a sharp decrease in carrying capacity, productivity and biomass production. Degradation may result from inadequate soil/plant/herd management, mostly common throughout overgrazing, insufficient weed and pest controls and lack of fertilization [11].

According to Kwon et al (2015)[15] there are about 1.1 Gha of degraded pasturelands ~~over~~across the world. In Brazil, the most comprehensive and exhaustive satellite imagery and geoprocessing analysis on pasturelands, the Atlas of Brazilian Pasturelands, ~~had~~ identified in 2017 that there were 95 Mha of degraded pasturelands – from a total of 190 Mha - with 40 Mha in severe level of degradation [5].

However, ranchers recognize only 12 Mha of pastures in poor condition on their properties, as stated in the last Agricultural Census conducted in 2017, which covered all the 5 million farms around the country [4].

Discrepancy between geoprocessing analysis and the recognition of the degraded conditions of their own pasturelands by the farmers is expected, and may have several causes, from the perspectives to interests [16], a complexity discussion not addressed here. For the purpose of this study is important to highlight the recognition of the degraded pastures by the ranchers is a crucial step to changing the adoption of more efficient and productive technologies [15].

The technical or empirical framework used by the ranchers reflect their beliefs, and knowledge and it is closer to the local experiences and practices. Focusing on what was declared in the census instead what was identified by GIS experts underestimates the area that could be recovered and so how large are the opportunities to the Brazilian livestock sector. On the other hand, preserving the decision processes in the allocation of production factors is in line with rancher perspective, providing subsidies for policies.

Commented [Rev8]: This is tagged onto the end of the sentence but it's not entire clear what is meant.

2.2 Technical Efficiency

Producers may be characterized as efficient if they have produced as much as possible with the inputs they have employed or whether they have produced the output at minimum cost.

Commented [Rev9]: One sentence paragraph. Should be combined elsewhere.

In our empirical application, we calculate efficiency by adopting the concept of technical efficiency [17,18]. Technical efficiency measures the distance of observed production to the potential maximum production. It may be interpreted as the additional output the farm could produce while using the current amount of inputs or, conversely, the potential reduction in input use that could be attained while producing the current output quantity. Formally, the concept may be expressed by the formula

$$TE = \frac{y}{\bar{y}} \leq 1 \quad (1)$$

where TE denotes technical efficiency, y represents the observed output and \bar{y} the potential (maximum) output. As y tends to the potential output \bar{y} , TE increases and the firm becomes more efficient. Low TE values mean that the observed production is far below the potential level, indicating an inefficient firm.

There are several approaches to estimating technical efficiency. We adopt the Stochastic Frontier Analysis, which proposes an econometric-based approach. In order to derive an econometric model, we first observe that potential output \bar{y} may be represented by the production function $f(\mathbf{x}, \boldsymbol{\beta})$, where \mathbf{x} is a vector of input quantities and $\boldsymbol{\beta}$ is a vector of parameters to be estimated. Equation (1) may be rewritten as

$$y = \bar{y}TE = f(\mathbf{x}, \boldsymbol{\beta})TE \quad (2)$$

The production model is usually linear in the log of when the variables are on a logarithmic scale (in our application, we adopt a Cobb-Douglas functional form), so the empirical model for farm i may be expressed as

$$\ln(y_i) = \ln f(\mathbf{x}_i, \boldsymbol{\beta}) + \ln(TE_i) = \ln f(\mathbf{x}_i, \boldsymbol{\beta}) - u_i \quad (3)$$

where $u_i \geq 0$ is a measure of technical inefficiency since $u_i = -\ln TE_i \approx 1 - TE_i$. We observe that

$$TE_i = -u_i \quad (4)$$

Our production function is specified as a Cobb-Douglas and our input vector \mathbf{x} includes capital, labour and land. Pasturelands are disaggregated in three components: degraded, natural and planted pasturelands. Such decomposition allows us to assess how technical efficiency is associated ~~with~~ each type of pastureland. In particular, we may evaluate how restoration of degraded pastureland may reduce technical inefficiency.

2.3 Stocking rates

The additional herd due to the recovery of degraded pasturelands is a function of area and productivity. As described in 2.1 the degraded pasture area was defined as declared by farmers. For the potential productivity, differently from previous studies that account for the biophysical potential – which threshold is defined by the potential of carrying capacity due to soil/biomass/herd relations [12,19]- we adopted as threshold the current stocking rate of non-degraded pasturelands.

Non-degraded pastures may be both native or planted pasturelands with different stocking rates, due to natural and management attributes. The estimated stocking rate for each one provides the current threshold that degraded pastureland could achieve wherever would be the biophysical potential, following the assumption of non-disruptive transformation limited to the actual inputs allocation, aligned with technical efficiency approach.

Again, the non-disruptive intensification considered here (1) is limited to the 12 M ha recognized by the producers and (2) supposes the increase is limited to average stocking rates currently observed and under the widespread technology.

These two hypotheses allow us to interpret our results as conservative estimates, since there would be room for the recovery of larger areas of degraded pastures and productivity gains due to technological advances. This reinforces the technical and economic viability of non-disruptive recovery actions and indicates that the implementation of these actions would not threaten to trigger the supposed rebound effect caused by the extraordinary intensification (the one that could potentially be reached by biophysical limits).

Stocking rate may be understood as the marginal impact of one additional unit of pastureland area, and therefore estimated as a regression coefficient. Estimations based standard regression models as OLS will suffer from a misspecification and the results of the model will be biased or inconsistent if the regressors, residuals or the dependent variable are spatially dependent, as described by Fisher & Wang 2011 [20]

Pastures are explicitly spatial variables, and testimonies of local landscape management. In order to estimate stocking rates we applied the Spatial Error Model which adequately treats spatial dependence in the error term that arises from unobservable latent variables that are spatially correlated. Formally, the SEM may be expressed by the formula

$$\begin{aligned} Y &= X\beta + \varepsilon \\ \varepsilon &= pW\varepsilon + u \end{aligned} \quad (5)$$

where Y denotes the dependent variable, X the matrix of independent variables, β the corresponding parameter vectors of X , ε error term, p the spatial scalar disturbance, W the

Commented [Rev10]: Awkward wording – need to correct

Commented [Rev11]: Another 1 sentence paragraph that need to be incorporated into another paragraph.

Commented [Rev12]: I'm not sure what the two "hypotheses" are as they haven't been clearly stated. Do you mean the two "assumptions" in the prior paragraph?

Commented [Rev13]: Noun needed prior to 'reinforces'. Not clear what "this" refers to.

spatial-weighting matrix and u the random error. The $pW\varepsilon$ is the autoregressive component in error term.

$$\text{Assuming } |p| < 1, \quad \varepsilon = (I - pW)^{-1}u \quad (6)$$

The equation (5) may be described as

$$Y = X\beta + (I - pW)^{-1}u \quad (7)$$

W is the row-standardised spatial weight matrix ~~W~~ built from an inverse of distance matrix between all sample municipalities (km).

$$W = \begin{bmatrix} 0 & \frac{1}{d_{1,2}} & \dots & \frac{1}{d_{1,j-1}} & \frac{1}{d_{1,j}} \\ \frac{1}{d_{2,1}} & 0 & \dots & \frac{1}{d_{2,j-1}} & \frac{1}{d_{2,j}} \\ \vdots & \vdots & \ddots & \vdots & \vdots \\ \frac{1}{d_{i-1,1}} & \frac{1}{d_{i-1,2}} & \dots & 0 & \frac{1}{d_{i-1,j}} \\ \frac{1}{d_{i,1}} & \frac{1}{d_{i,2}} & \dots & \frac{1}{d_{i,j-1}} & 0 \end{bmatrix} \quad (8)$$

where $d_{i,j}$ = distance between municipality i and municipality j

Given this final specification we grouped all 5,563 the municipalities into 8 regions according to predominate biome, considering the predominant biome was the one with the largest share of the municipality area. For each region we ran a specific Spatial Error Model Regression with Y been the cattle herd, and X a matrix with degraded, native and planted area and controls.

3 Additional Herd scenarios were built: (LIS) Low Intensification Scenario- if all degraded pasturelands were recovery to achieve the same stocking rate estimated to native pasturelands, (HIS) High Intensification Scenario - if all the degraded pasturelands were recover~~ed~~**edy** to achieve the same stocking rate estimated ~~for~~**to** planted pasturelands, and (FCS) Forest Code Compliance - degraded pastures are allocated for forest restoration in order to reach the minimum area necessary to cover the deficit of Legal Reserves as mandated by Forest Code Law. Surplus degraded pastures, if any, were allocated to be recover to achieve the same stocking rate estimated to planted pasturelands.

3. Results and Discussion

3.1 Efficiency, Stocking Rates and Additional herd on degraded pastures

Frontier model of stochastic production based on Cobb-Douglas production function shows that Brazilian livestock currently has an average technical efficiency of 0.81 (Figure 1), ~~which~~**ed**. ~~This~~**is** means that, on average, current production corresponds to 81% of the potential level of production given the factors currently employed. Incomes could increase 19% without additional inputs, including land.

Among the factors of production analysed, land (native, planted and degraded pastures), capital (tractors and agricultural machinery and equipment), labor (~~employed persons~~) and technical, and biophysical controls, the degraded pasture showed the lowest marginal productivity (-0.041), making it being the main factor of diseconomy of scale in livestock production. The analysis also shows that although there are levels of efficiency that vary between the 27 States, degraded pasture reduces the average yield of livestock activity in the municipality, regardless of the region, and ~~is~~ associated with municipalities with higher levels of technical inefficiency.

Figure 1 – Technical Efficiency of Livestock in Brazil

A collection of ~~Several~~ studies carried out in Brazil at the farm level indicates that the recovery of pastures could increase productivity between 2 to 10 times, depending ~~basically~~ on the level of initial degradation and the recovery techniques used, which would lead to an increase of up to 8 times the producer's incomes and up to 3 times its net profit [19]. From a scale perspective, other studies have shown that the country could increase its total livestock production by up to 150%, without needing any additional area, using only the intensification of the 100 Mha of pastures already available [11,12,13,14].

However, an effort of this magnitude is doubtful to be possible. Focusing on the 12 million hectares of degraded pastures already recognized by producers, may be ~~a~~ more plausible strategy, especially because the main barrier - the perception of the problem by the ranchers themselves - would already be overcome.

Our results indicate that planted pasture, in all regions, represents the highest marginal impact in the to-increase the stocking rate of herd—highest stocking rate—, which is the double or even the triple of stocking rate found for native pastures. It corroborates studies carried out in the field or on experimental farms which have reported that native pastures have carrying capacity ranging from 0.2 to 0.5/heads/ha due to the low digestible biomass production, low palatability or high biodiversity, which imposes more time for selecting feed and larger areas for foraging [11].

Figure 2 – Regions and stocking rates estimated by SEM models. Stocking rates are measured as additional head per additional hectare of planted, native and degraded pastures. *** is significant at 1%, ** at 5% and * at 10%.

Results indicate that an increase in the area of planted pastures is associated with a higher stocking rate when compared to variations in other types of pasture, reinforcing the knowledge built over the last decades, especially by Embrapa. It is important to note that a marginal increase in the degraded pasture area has a negative impact on the average stocking rate in the main growth regions of the Brazilian herd. This result reinforces the need to recover degraded pastures in order to promote greater intensification of livestock, thus reducing the pressure for deforestation, and is in line with efficiency outputs.

Commented [Rev14]: Need to explain to international reader what Embrapa is – e.g. the Brazilian government agricultural agency.

The recovery of degraded pastures would increase the cattle herd to 16.9 million heads in the High Intensification Scenario –equivalent to the current size of Indonesia’s herd. Contributions of Center-Southern Cerrado, Amazon and Southern Atlantic Forest (EAF) would range respectively from 36 to 44%, 27% to 41% and 9% to 13%.

Most impressive is even under the FCS, which prioritizes the allocation of degraded pastures for forest restoration, the increase of the herd in the surplus areas could reach 9.1 M heads.

Figure 3 Additional herd by region in the 3 Scenarios.

This does not necessarily mean that it would be possible to cover all Legal Reserve deficits and still produce more **cattle** with the degraded pastures. There are 831 municipalities where the degraded pasture area is insufficient to cover the deficit, even if fully allocated for forest restoration, and, in these cases, no additional herd would be counted.

Brazilian legislation establishes that Legal Reserves deficits registered in one region can be compensated in other regions. However, there are well-delimited restrictions, especially within the Biome and the administrative divisions of States. On the other hand, the Brazilian legislation is clear in stating that land use planning is mainly attributed to municipalities, so that the approach of focusing on the municipal level does not exhaust the possibilities of allocating pastureland for production or forest restoration forestation but sounds well suited for the level of discussion defended here

Natural areas and restored forests play crucial roles for the economy and competitiveness supplying the inputs for which there are neither economically or technically viable large-scale substitutes, such as rainwater irrigation, soil and water conservation, pollinator shelters and climate stability [21, 22]. In addition to this economy, **which is exclusive to the standing forest - and which encompasses ecosystem services for agriculture as a whole - other conventional explorations like cattle raising in native pasturelands may currently play a greater role in the generation of direct income and jobs [4], and therefore has greater appeal, especially in countries with great social inequality such as Brazil.**

Commented [Rev15]: Sentence is way too long and hard to follow. Need to reword to make this point clearer and more succinct.

Deforestation and degradation of pastures are phenomena that have **fed edbacked** into the process of advancing the frontier in Brazil [6,11, 23]. Differently, forest restoration and pasture recovery have surged as the two pillars of the global food security strategy, which intelligence lies in transforming degraded areas as the new frontier [24]. The simultaneous

expansion of degraded areas for agricultural production and reforestation revitalizes the rural economy and rehabilitates the landscape to provide the environmental services that foster production itself, and it is not new for Brazil [25]. The recovery of degraded areas for forest restoration and livestock intensification are feasible in Brazil

Commented [Rev16]: Citations needed here.

It is important to note that degraded pastures and deficits are concentrated in specific regions greatly facilitates the adoption of these strategies at the regional level. The regional concentration allows the focus of public and private investments to leverage the recovery of pastures. To illustrate this point, consider that a policy to recover 10 M ha of degraded pastures in livestock establishments was defined in 10 stages of 1 M ha, in which municipalities are being incorporated according to the magnitude of the degraded pasture area. The first million of the first recovery stage would be concentrated in only 15 municipalities, the second in 27 and the third million in another 41 municipalities, in such a way that the recovery of 25% of degraded pastures could be directed at contemplating 1% of Brazilian municipalities.

Commented [Rev17]: Confusing sentence.

Figure 4: Accumulated concentration of degraded pastures in cattle farms according to the municipalities. (A) the 15 municipalities with the largest area of degraded pastures and which together hold 1 M ha of degraded pastures potentially increasing 1.4 M heads. (B) the 42 municipalities that together hold 2 M ha and would increase 2.79 M heads, and (C) the 83 municipalities that together hold 3 M ha and would increase 4.02 M heads.

It opens a new frontier in Brazil based on degraded pasturelands. Even for pastures in good condition, current productivity is well below its potential, dampening the effect of Jevons VIII, which provides for a greater risk of resource depletion just when it becomes more efficient. The race for extraordinary intensification and its possible rebound effect on new deforestation are avoided.

Commented [Rev18]: What is Jevons VIII?

Commented [Rev19]: This paragraph is redundant with points made elsewhere and could be cut.

3.2 Rural Credit and Technical Assistance

Commented [Rev20]: There's no discussion of the broader literature outside Brazil.

Our simulations indicate that increasing the stocking rate through the recovery of degraded pastures can bring simultaneous environmental and economic gains. The intensification of livestock represents a great saving of land immobilized in the productive system, open a new frontier.

Commented [Rev21]: Again this is a really generally statement that is made elsewhere in the paper.

There are several drivers that act as barriers to the adoption of degraded pasture recovery practices, among them most important are lack on insufficient Technical Assistance (ATER) and Rural Credit [16, 26,27,28]. On fact, according the latest Agricultural Census only 1 in 10 cattle ranchers has intensive and frequent technical assistance. More than 77% of farms do not have technical assistance. There are 131 million hectares of pasturelands without professional orientation, a chronic problem that is not restricted to small productions, since the percentage unattended by ATER does not differ statistically among the different groups of production value, which raises the challenge far beyond the land issue [4].

Commented [Rev22]: Degraded pasture recovery practices have not been described yet. This should be part of the introduction.

Lack of investments is ~~even worse~~ ~~an even larger obstacle to pasture recovery~~. In the past 12 years, the official rural financial credit system has resources equivalent to just R \$ 12 / ha / year in pastures iv. Even though a 5-year pasture renewal cycle is considered, adding all the average investment credit values under the following headings: fertilization; soil correction and protection; formation, reform and recovery of pastures, the sum would not reach the lowest R \$ 62 / ha / five-year period. A value 30 times lower than that recommended by Embrapa. We could not expect a different reality in the productivity gains of the sector with such low levels of investment [29].

It is true that these values must be analysed with caution, because the items of products by activity (e.g. culture or goods and services) financed by Rural Credit are quite generic. Even so, for the period from 2015 to 2019 (in which there is consistency in the disclosure of the Rural Credit Data Matrix for the amount of area, investment and costs), we estimated at constant values of 2019 equivalent à R \$ 36.4 billion diluted in an area equivalent to 137.6 million hectares of pasture, or an average of R \$ 264 / ha / year [29]. ~~This is~~ is still 7 times less than what is necessary to reverse the unproductive situation of Brazilian livestock [11, 19].

Commented [Rev23]: This sentence is hard to follow.

The low level of rural credit contracted by livestock ranchers - although with considerable growth in the last 5 years - is due to very well-known drivers. On the demand side, (1) high liquidity of the activity, which allows anticipating revenues, (2) less willingness to bank indebtedness, compared to farmers, behaviour historically evidenced by time series in Agricultural Census and Central Bank, (3) lower levels of technical and financial planning, also compared to agriculture, which implies less ability to compete with non-targeted resources, especially provided by the Safra Plan; From the supply point of view, (1) greater difficulty for the creditor tracing application of the resource - especially in breeding farms, (2) lower levels of guarantee, especially in the North region, where there are serious problems in land title and tenure.

Commented [Rev24]: Unclear.

Commented [Rev25]: Not clear.

Commented [Rev26]: What is the Safra plan?

Commented [Rev27]: This whole paragraph needs to be rewritten in a way that is clearer to the reader.

~~We evaluated~~ Our results show that livestock efficiency is positively associated with the increase in ATER and rural credit. On average, municipalities with less access to the resources of the Safra Plan and smaller areas with ATER services are those with lower levels of technical efficiency. Municipalities with higher amounts of credit and with a greater proportion of areas with ATER have, on average, greater technical efficiency, as well as less dispersion of the efficiency index. This result is not surprising, since greater access to financial and technical resources should also improve the management quality of agricultural establishments.

Commented [Rev28]: Refer to Figure 5.

Commented [Rev29]: This says the same thing as the previous sentence. Delete one of them.

Figure 5: Livestock efficiency positively associated with the increase in ATER and rural credit.

Commented [Rev30]: But there are lots of farms that are efficient with little credit. There's just a wider range in efficiency.

Rural Credit is also crucial. From 2013 to 2019, 13 different credit lines were granted, equivalent to U\$ 3.45 billion in credits for pasture, of which US 1.4 billion financed by ABC. The 1600 municipalities in which ABC was responsible for at least 20% of the total credit granted for pastures showed a level of technical efficiency statistically superior to the others, regardless of the region and the area assisted by ATER itself [29].

Commented [Rev31]: Acronym not spelled out.

One of the hypotheses for this superior efficiency is precisely the fact that the ABC is the only one of the 13 lines that explicitly subject the concession of the resource to the reform of pastures, while the others do not bring this requirement, and can simply afford the production without innovation or even finance the formation of pastures over areas of native vegetation, which, in one case or another, reinforces cultural inertia instead of promoting reform. And ABC has not reached some priority cities. The 11 municipalities estimated here that concentrate 10% of degraded pastures in Brazil take only 3.5% of the credits contracted by ABC. The gap opens as the 40 municipalities that hold 25% of degraded pastures take only 7.5% of the ABC while the 152 municipalities that concentrate 50% of the degraded pastures access 20% of the ABC.

The first step in improving the efficiency of the ABC Program is to direct the Program's resources to priority municipalities. Respecting budgetary constraints and the levels of subsidies, subsidies or spread differences, it would be via the displacement of resources from the Rural Savings and Constitutional Funds that currently encourage financing unrelated to specific programs. We estimate that, in the last 7 years, a total of US \$ 845 M from Rural Savings (under subsidized interest rates) and another U\$ 116 M from constitutional funds financed pastures and intensive soil management in livestock via non-specific program, and that they could be directed to the ABC, thus ensuring greater efficiency in the conversion of the resource to recover pastures and improve production and revenue for the producer. Annually, it would cost be, on average, an additional U\$ 137 M, which would represent an increase of 70% for ABC resources for pastures and intensification of the soil for livestock.

Commented [Rev32]: Unclear sentence.

Commented [Rev33]: Run on sentence.

Figure 6: Sources and Credit Programs which have financed recovery of pasturelands in Brazil between 2013 and 2019. Values in US M.

It should also be noted that the rural credit subsidy, extended beyond the ABC Program, should be justified as a counterpart in terms of positive environmental externalities. Clearly, credit directed to the recovery of pastures meets this criterion, since it involves productivity gains and reduced pressure for the incorporation of new lands. This argument reinforces the need to redirect subsidized rural credit to finance the recovery of pastures, to the detriment of financing the cost where environmental externalities are often absent and credit acts as public financing of private margins with no social counterpart.

Commented [Rev34]: I'm not sure what you're trying to say here.

In the case of livestock, technologies to increase the productivity of pastures are generally simple and are already widespread in many Brazilian regions, which greatly reduces the risks of technological change driven by credit. Specifically for degraded pastures, the ~~and~~ recovery already show productivity leaps in the first year, generating working capital that can sustain the producer during the grace period, in such a way that neither the grace period nor the discharge terms would need to be adjusted.

Legal conditions for granting credit are essential to reinforce the application of the resource in an efficient and healthy manner or to prevent dilution of the application in illegal horizontal expansion. The positive effect of Resolution 3545 issued by the Central Bank in 2008 that conditioned subsidized credit in the Amazon to compliance with land tenure and Forest Code, shows that the country is prepared for this [28].

But rural credit is not the only and not always the main source of financing for livestock. Since 1970, the share of credit made available by the National Rural Credit System in total livestock expenses has varied from 22 to 47% (95% confidence to average) [30, 31, 32, 33, 34]. Thus, bank loans with market interest rates and the use of own individual resources are

essential sources for financing activity in Brazil. We estimate that in 2017 this amount was no less than U\$ 18 Bn [29], which is the balance obtained when comparing the ~~costing and~~ investment expenses of livestock establishments in 2017 with funds contracted for livestock via the Safra Plan that same year.

If rural credit should encourage ~~livestock to~~ steer pasture recovery by recognizing the value of externalities and justifying subsidies. ~~P-~~private capital without subsidies can pull good commercial practices that value producers and their products into the chain. The enforceability in complying with legislation and valuing good practices - made, for example, in complete product tracking - is the effective way to finance changes downstream in the production process, promoting the capitalization of the producer increasing its competitiveness

Commented [Rev35]: Livestock don't steer pasture recovery. Need to reword this sentence.

Livestock ~~producers~~ that aspires to growing markets must invest in productivity, and the reform of degraded pastures is the great asset of Brazil, its new frontier. ~~Not only because it represents a great saving of land immobilized in the productive system, but also because it is able to leverage one of the great comparative advantages: grass-fed systems.~~ Compared to American and European grain fed or confinement systems, the Brazilian grass fed grazing system has many advantages that have been valued in the international market (1) better sanitary conditions - less risk of contamination by pathogens and development of diseases for animals and humans, (2) less use of medicines, (3) greater animal comfort and (4) less competition for access to chemical inputs and raw materials, including the grains that can be used for human consumption instead of bovine feed [35].

Commented [Rev36]: It's not the livestock that aspires to growing markets

Commented [Rev37]: This exact verbiage of the new frontier is repeated multiple times. Reduce the repetition of generalities and use the space to clarify complex topics.

Commented [Rev38]: Incomplete sentence.

~~Degraded pasturelands as the new frontier and could enhance those advantages.~~

4. Conclusions

Commented [Rev39]: This section reads more like a summary or abstract than conclusions. You don't need to repeat results here. Just focus on a couple of conclusions that are interpreted in the broader literature, not just the Brazilian case.

The inefficiency of Brazilian livestock is very high, around 28%. The degraded pasturelands present the main factor on diseconomy of scale and could become the new frontier. The recovery of degraded pastures, starting with 12 M ha recognized by the own ranchers, is a huge opportunity, with widespread technology and high impact to increase the productivity and competitiveness of Brazilian livestock in the short term. Additionally, it would help the sector to detach itself from the label of the largest vector of deforestation.

These degraded pastures are highly concentrated in a few regions, which would make investment programs in recovery less susceptible to the transition costs inherent in the capillarity of credit programs or even ATER, thus ensuring greater effectiveness in the productive conversion of these pastures. 1% of Brazilian municipalities concentrate 34% of degraded pastures

Rural Credit can have a significant impact ~~o~~in reducing this inefficiency, as they are important bottlenecks for this economic and productive gain. For example, 3 out of 4 hectares of rural establishments with livestock do not have an ATER. On average, livestock farms invest 7 to 30 times less than necessary to recovery pastures. On the other hand, Rural Credit finances only U\$ 1 in every U\$ 4 invested in livestock, and it is important to redirect the costing credit for investment.

The ABC Program, especially the subprogram “Recovery of Degraded Pastures” must be broadly expanded, and one can begin by redirecting resources from Rural Savings with controlled interest, and currently available to promote pastures without being linked to specific programs.

Acknowledgments

The authors express their deep gratitude for the valuable discussions and voluntary contributions received from Joaquim Levy, João Adrien, Leia Harfuch, Leonardo Fleck, Mariane Crespolini dos Santos and Miguel Calmon. Authors are also grateful to Berta Pinheiro, Carolina Genin, Marcelo Matsumoto, Leonardo Barbosa, Talita Esturba and Viviane Romeiro for their volunteer scientific and technical support.

Funding Statement

This research was originally funded by WRI

Data Accessibility

Data are available at doi:10.5061/dryad.76hdr7sv8
Supplemental Material available

Competing Interests

Authors declared no competing interests section.

Authors' Contributions

Rafael Feltran-Barbieri (RFB) carried out literature review, database collection, developed spatial regression models scripts and analysis, led the design of the study and led the manuscript. Jose Gustavo Feres (JGF) contributed to literature review, database collection, developed stochastic frontier model, participated in the design of the study and contributed to the manuscript

References

1. United States Department of Agriculture USDA. 2020. Livestock and Poultry: World markets and trade. Washington, Foreign Agricultural Service, USDA. See https://downloads.usda.library.cornell.edu/usda-esmis/files/73666448x/sb397r25n/0z709c25b/livestock_poultry.pdf (accessed on 21 July 2020)
2. Associação Brasileira das Indústrias Exportadoras de Carne ABIEC. 2020. Exportações por parceiro. See <http://abiec.com.br/exportacoes/> (accessed on 31 July 2020)
3. Centro de Estudos Avançados em Economia Aplicada da ESALQ/USP, Confederação Nacional da Agricultura – Cepea/CNA. 2019. PIB do Agronegócio Brasileiro. Piracicaba/Brasília CEPA/CNA. See <https://www.cepea.esalq.usp.br/br/pib-do-agronegocio-brasileiro.aspx> (accessed on 4 August 2020)
4. IBGE Instituto Brasileiro de Geografia e Estatística 2019. Censo Agropecuário 2017: resultados definitivos. Rio de Janeiro IBGE. See <https://sidra.ibge.gov.br/pesquisa/censo-agropecuario/censo-agropecuario-2017> (accessed 18 June 2020)
5. Laboratório de Processamento de Imagens e Geoprocessamento LAPIG. 2020. Atlas das pastagens brasileiras. Goiânia: LAPIG/UFG. See <https://www.lapig.iesa.ufg.br/lapig/index.php/produtos/atlas-digital-das-pastagens-brasileiras> (accessed on 12 October 2020)
6. Parente L, Mesquita V, Miziara, F, Baumman, L, Ferreira L. 2019. Assessing the pasturelands and livestock dynamics in Brazil, from 1985 to 2017: a novel approach based on high spatial resolution imagery and Google Earth Engine cloud computing. **232**, 111303 (doi:10.1016/j.rse.2019.111301)
7. Mapbiomas Projeto de Mapeamento Anual da Cobertura e Uso do Solo do Brasil 2019. Cobertura e Uso do Solo. São Paulo: Mapbiomas versão 5.0. See <https://plataforma.mapbiomas.org/map#coverage> (accessed on 12 October 2020)
8. Azevedo TR, Rosa MR, Shimbo JZ, Martin EV, Oliveira MG. 2020. Relatório Anual de Desmatamento 2019. São Paulo: Mapbiomas. See <https://s3.amazonaws.com/alerta.mapbiomas.org/relatorios/MBI-relatorio-desmatamento-2019-FINAL5.pdf> (accessed 9 October 2020)
9. Coalizão Brasil Clima, Florestas e Agricultura. 2020. Ações para a queda rápida do desmatamento. See http://coalizaobr.com.br/boletins/pdf/Acoes_para_a_queda_rapida_do_desmatamento-CoalizacaoBrasil.pdf (accessed on 3 October 2020)
10. Martha Jr GB, Alves E, Contini E. 2012. Land-saving approaches and beef production growth in Brazil. **110**, 173-177 (doi:10.1016/j.agsy.2012.03.001)

Field Code Changed

Field Code Changed

Field Code Changed

11. Dias-Filho MB. 2014. *Diagnóstica das pastagens no Brasil*. Belem: Embrapa Amazônia Oriental. Documentos 402. See <https://www.infoteca.cnptia.embrapa.br/bitstream/doc/986147/1/DOC402.pdf> (accessed on 2 May 2020)
12. Strassburg BBN et al. 2017. Moment of truth for the Cerrado hotspot. **1**, 0099 (doi: 10.1038/s41559-017-0099)
13. Oliveira Filho R et al. 2017. Sustainable intensification of Brazilian livestock production through optimized pasture restoration. **153**, 201-211. (doi:10.1016/j.agsy.2017.02.001)
14. Van Zanten HHE et al. 2016. Global food supply, land use efficiency of livestock systems. **21**, 747-758 (doi:10.1007/s11367-015-0944-1)
15. Kwon HY, Nkonya E, Johnson T, Graw V, Kato E, Kihui E. 2015. Global estimates of the impacts of grassland degradation on livestock productivity from 2001 to 2011. In Nkonya E, Mirzabaev A, von Braun J (eds). *Economics of land degradation and improvements: a global assessment for sustainable development*. Cham: Springer
16. Latawiec AE et al. 2017. Improving land management in Brazil: a perspective from producers. **240**, 276-286. (doi: 10.1016/j.agee.2017.01.043)
17. Farrell MJ. 1957. The measurement of productive efficiency. *J. R. Stat. Soc.* **120**(3), 253-281
18. Belotti F, Daidone S, Ilandi G, Atella V. 2013. Stochastic frontier analysis using Stata. **13**(4), 719-758. (doi: 10.1177/1536867X1301300404)
19. Assad ED et al. 2019. Papel do Plano ABC e do Planaveg na adaptação da agricultura e da pecuária às mudanças climáticas. São Paulo: WRI Brasil. See <https://wribrasil.org.br/pt/publicacoes/papel-do-plano-abc-e-do-planaveg-na-adaptacao-da-agricultura-e-da-pecuaria-mudancas> (accessed on 19 April 2020)
20. Fisher MM, Wang J. 2011. *Spatial Data Analysis: Models, Methods and Techniques*. Dordrecht: Springer
SpringerBriefs in Regional Science
21. Barbut M., Alexander S. 2016. Land degradation as a security threat amplifier: the new global frontline. In Chabay I, Frick M, Helgeson J. (eds). *Land Restoration: reclaiming landscapes for a sustainable future*. Elsevier/Academic Press
22. Chazdon RL. 2008. Beyond deforestation: restoring forests and ecosystem services on degraded lands. **320**, 1458-1460 (doi: 10.1126/science.1155365)
23. Gibbs HK, Salmon JM. 2014. Mapping the world's degraded lands. **5**, 12-21. (doi: 10.1016/j.apgeog.2014.11.024)
24. Webb NP, Marshall NA, Stringer LC, Reed MS, Chappell A, Herrick JE. 2017. Land degradation and climate change: building climate resilience in agriculture. **15**(8), 450-459 (doi: 10.1002/fee.1530)
25. Nepstad DC, Uhl C, Serrao EAS. 1991. Recuperation of a degraded Amazonian landscape: forest recovery and agricultural restoration. *Ambio* **20** (6), 248-255
26. Harfuch L, Palauro G, Zambianco W. 2016. Análise econômica de projetos de investimento para expansão da produção pecuária. São Paulo: INPUT Agroicone/ Moore. See https://www.inputbrasil.org/wp-content/uploads/2016/11/An%C3%A1lise-econ%C3%B4mica-de-projetos-de-investimentos-para-expans%C3%A3o-da-produ%C3%A7%C3%A3o-pecu%C3%A1ria_Agroicone_INPUT.pdf (accessed on 18 May 2020)
27. Harfuch L, Nassar AM, Zambianco WM, Gurgel AC. 2016. Modelling beef and dairy sector's productivities and their effects on land use change in Brazil. **54**(2), 281-324 (doi: 10.1590/1234.56781806-947900540205)
28. Assunção J, Gandour C, Rocha R, Rocha R. 2020. The effect of rural credit on deforestation: evidence from the Brazilian Amazon. **130**, 290-330 (doi: 10.1093/ej/uezo60)
29. Banco Central do Brasil. 2020. Matriz de Dados do Crédito Rural. Série temporal de crédito concedido por município, ano de emissão, produto, programa e sub-programa, from e-sic. See <https://esic.cgu.gov.br/sistema/site/index.aspx> (accessed on 2 February 2020)
30. Fundação Instituto Brasileiro de Geografia e Estatística IBGE. 1975. Censo Agropecuário 1970, VIII Recenseamento Geral -1970. Rio de Janeiro: FIBGE. Série Nacional Volume 3. See https://biblioteca.ibge.gov.br/visualizacao/periodicos/45/ca_1970_v3_br.pdf (accessed on 6 July 2020)
31. Fundação Instituto Brasileiro de Geografia e Estatística IBGE. 1979. Censo Agropecuário 1975. Censos Econômicos de 1975, Brasil. Rio de Janeiro: FIBGE. Série Nacional Volume 1. See https://servicodados.ibge.gov.br/Download/Download.ashx?http=1&u=biblioteca.ibge.gov.br/visualizacao/periodicos/243/agro_1975_v1_br.pdf (accessed on 6 July 2020)
32. Fundação Instituto Brasileiro de Geografia e Estatística IBGE. 1984. Censo Agropecuário 1980. IX Recenseamento Geral do Brasil 1980. Rio de Janeiro: FIBGE, Série Nacional Volume 2, Tomo 3, Numero 1. See https://biblioteca.ibge.gov.br/visualizacao/periodicos/46/ca_1980_v2_t3_n1_br.pdf (accessed on 6 July 2020)
33. Fundação Instituto Brasileiro de Geografia e Estatística IBGE. 1991. Censo Agropecuário
1985. Censos Econômicos de 1985. Rio de Janeiro: FIBGE. Volume 1. See https://biblioteca.ibge.gov.br/visualizacao/periodicos/47/ca_1985_n1_br.pdf (accessed on 6 July 2020)
34. Instituto Brasileiro de Geografia e Estatística IBGE. 1998. Censo Agropecuário 1995-1996. Rio de Janeiro: IBGE Volume 1. See https://biblioteca.ibge.gov.br/visualizacao/periodicos/48/agro_1995_1996_n1_br.pdf (accessed on 6 July 2020)
35. Stone Barns Center for Food and Agriculture. 2017. Back to grass: the market potential for US grassfed beef. ND:Stone Barns, Bonterra SLM. See https://www.stonebarnscenter.org/wp-content/uploads/2017/10/Grassfed_Full_v2.pdf (accessed 03 August 2020)
36. Instituto Nacional de Colonização e Reforma Agrária INCRA. 2020. Modulo Fiscal. See http://www.incra.gov.br/media/docs/indices_basicos_2013_por_municipio.pdf (accessed on 31 August 2020)
32. Food and Agriculture Organization FAO UN. 2016. World Programme for the Census of Agriculture 2020. Methodology. See <http://www.fao.org/world-census-agriculture/methodology/en/> (accessed on 9 July 2020)
37. United Nations Statistic Division. 2007. International Standard Industrial Classification of all Economic Activities – ISIC. Revision 4. See <https://unstats.un.org/unsd/classifications/Econ/isic> (accessed on 9 July 2020)
38. Instituto Brasileiro de Geografia e Estatística IBGE. 2019. Código dos Municípios IBGE. See <https://www.ibge.gov.br/explica/codigos-dos-municipios.php> (accessed on 19 September 2020)
39. Zomer RJ, Trabucco A, Bossio DA, Verçjot LV. 2008. Climate change mitigation: A spatial analysis of global land suitability for clean development mechanism afforestation and reforestation. **126**, 67-80 (doi:10.1016/j.agee.2008.01.014) See for database <https://cgiarcsi.community/2019/01/24/global-aridity-index-and-potential-evapotranspiration-climate-database-v2/> (accessed on 15 October 2020)
40. Instituto Brasileiro de Geografia e Estatística IBGE 2020. Malha municipal. See <https://www.ibge.gov.br/geociencias/organizacao-do-territorio/15774-malhas.html?=&t=downloads> (accessed on 19 September 2020)

Appendix C

We are very grateful for the valuable reviews, comments, and suggestions from the anonymous reviewers. A complete and detailed response is present in the new manuscript, track changes version. In this document we make only short responses.

Reviewer: 1

Comments to the Author(s)

An important topic in the context of the Brazilian NDC and its targets related to land use. Appropriate timing. The paper needs a few improvements and a more detailed list can be found in the attached file.

Thank you very much. Hope the changes have met your main expectations

Reviewer: 2

Comments to the Author(s)

The authors make the important point that pasture productivity could be improved in Brazil, which would hopefully decrease agricultural pressure on intact forest. I agree that this is an important message but the manuscript will need substantial revisions (detailed below) to be suitable for publication in Royal Society Open Science and accessible to a broad audience. I also made detailed edits on the manuscript.

Thank you very much. We did substantial review

1. The authors never explain early on what improving pasture productivity entails and this is important to understanding later portions of the manuscript. I think they mean fertilizing the pastures, moving cattle more frequently, and better animal management generally, but the specific practices need to be clearly described. A related important point is that the authors seem to use pasture restoration and recovery interchangeably, which is misleading. It seems that they are talking about improving cattle productivity (the term I suggest they use), rather than improving habitat quality in pastures. There is an extensive literature on silvopastoral systems (e.g., Murgueitio et al., 2011; Calle et al., 2013; Calle, 2020), which aims to do both, and I strongly recommend that the authors consider this literature and what might be achievable in Brazil. Regardless the authors need to clarify their use of terminology. What they are describing is not "restoration" (Gann et al., 2019).

Agreed. We have included 5 paragraphs reviewing the literature on the subject, highlighting the approach taken in the manuscript

2. The writing was hard to follow at multiple places in the article for a number of reasons. The authors repeat a few generalities many times (e.g. increasing pasture productivity is the new frontier), but then jam multiple technical terms into a single sentence without much background for the general reader. I have noted many cases of this in the paper. The authors alternate between some short paragraphs (just one sentence) and then very long sentences with multiple clauses. I note several that I found unclear. More generally, there are many run on sentences, missing words, and English grammar errors that made

the paper hard to follow. Once the other revisions are completed, the paper should be reviewed by a native English speaker.

Agreed. We restructured the text, detailing the concepts used, and tried to increase the linkage between the messages

3. The authors refer almost exclusively to Brazil and references from Brazil in this paper. I realize that Brazil is a large and complex country, but for an international journal the authors should consider how the Brazilian experience compares to other systems and countries. For example, there is an extensive literature on sustainable cattle ranching in Colombia. The authors also assume that various laws and organizations in Brazil (e.g. Embrapa, Safra Plan, ABC) are known to readers. Certainly, some readers of this article will know those terms, but for a general journal this knowledge should not be assumed and acronyms should always be spelled out. The authors assume a lot of knowledge about Brazilian agricultural credit.

Agreed. We included an introductory section contextualizing the problem of degraded areas in the world food production, the recommendations pointed out by international organisations such as FAO, OECD, IPCC and literature, and tried to insert Brazil in this context, creating a new and separate section for it.

We have included a specific section for the technical description of productivity and, throughout the manuscript, we have dialogued with other world experiences.

We reduced the scenarios used from 3 to 2, believing to give more clarity to the message, and gave more focus to the differences between forest restoration and pasture recovery. With all this, we practically doubled the bibliographical references

4. As noted below, multiple figures are lacking sufficiently descriptive captions.

A few more specific comments in addition to the many directly on the manuscript.

Title. It's not clear to the reader what the degraded pastures in Brazil are the new frontier for. They could be the new frontier for soybean growth given that the title is vague. The title needs to better communicate the main message of the paper.

Figure 1. A much more detailed caption is needed so the figure can be read and understood separately from the paper. This is true of several figures (e.g. Fig. 3 and 6 also).

Figure 3. The line for the y-axis is missing and a more descriptive caption is needed.

Figure 4. Is too small to be legible.

Agreed. We redo figures and eliminate some graphs. We included a detailed table of results and added and uploaded all the data used.